# The heart is a resident tissue for hematopoietic stem and progenitor cells in zebrafish

Dorothee Bornhorst [1,2], Amulya V. Hejjaji [1,2,9], Lena Steuter[1,2,9], Nicole M. Woodhead[3], Paul Maier[4], Alessandra Gentile [5,8], Alice Alhajkadour [3], Octavia Santis Larrain [3], Michael Weber [4], Khrievono Kikhi [6], Stefan Guenther [7], Jan Huisken[4], Owen J. Tamplin [3], Didier Y. R. Stainier [5] & Felix Gunawan [1,2] ✉

The contribution of endocardial cells (EdCs) to the hematopoietic lineages has been strongly debated. Here, we provide evidence that in zebrafish, the endocardium gives rise to and maintains a stable population of hematopoietic cells. Using single-cell sequencing, we identify an endocardial subpopulation expressing enriched levels of hematopoietic-promoting genes. High-resolution microscopy and photoconversion tracing experiments uncover hematopoietic cells, mainly hematopoietic stem and progenitor cells (HSPCs)/ megakaryocyte-erythroid precursors (MEPs), derived from EdCs as well as the dorsal aorta stably attached to the endocardium. Emergence of HSPCs/MEPs in hearts cultured ex vivo without external hematopoietic sources, as well as longitudinal imaging of the beating heart using light sheet microscopy, support endocardial contribution to hematopoiesis. Maintenance of these hematopoietic cells depends on the adhesion factors Integrin α4 and Vcam1 but is at least partly independent of cardiac trabeculation or shear stress. Finally, blocking primitive erythropoiesis increases cardiac-residing hematopoietic cells, suggesting that the endocardium is a hematopoietic reservoir. Altogether, these studies uncover the endocardium as a resident tissue for HSPCs/MEPs and a de novo source of hematopoietic cells.

A specialized subset of endothelial cells undergoes endothelial-to-hematopoietic transition (EHT) to give rise to hematopoietic stem and progenitor cells (HSPCs)[1–5]. EHT occurs in 1-3% of endothelial cells in the aorta-gonad-mesonephros (AGM) in mammals, the first vascular source of hematopoiesis[1–5]. These hemogenic endothelial cells upregulate factors critical to promote hematopoiesis, including the transcription factors Runx1, Gata2, and cMyb, the receptor Integrin αIIb/CD41, and members of the Notch, BMP, and Wnt signaling pathways[6–8]. The HSPCs then migrate into intermediary niches where the hematopoietic population expands before residing in their final

[1]Institute of Cell Biology, Faculty of Medicine, University of Münster, Münster 48149, Germany. [2]'Cells-in-Motion' Interfaculty Center, University of Münster, Münster 48149, Germany. [3]Department of Cell and Regenerative Biology, School of Medicine and Public Health, University of Wisconsin-Madison, Madison, WI 53705, USA. [4]Multiscale Biology, Faculty of Biology and Psychology, Georg-August-University Göttingen, Göttingen 37077, Germany. [5]Department of Developmental Genetics, Max Planck Institute for Heart and Lung Research (MPI-HLR), Bad Nauheim 61231, Germany. [6]Flow Cytometry and Cell Sorting Core Facility, MPI-HLR, Bad Nauheim 61231, Germany. [7]Deep Sequencing Platform, MPI-HLR, Bad Nauheim 61231, Germany. [8]Present address: Centre for Developmental Neurobiology, Institute of Psychiatry, Psychology and Neuroscience, King's College London, London, UK. [9]These authors contributed equally: Amulya V. Hejjaji, Lena Steuter. ✉e-mail: felix.gunawan@ukmuenster.de

niche, the bone marrow in mammals. HSPCs constitute only ~1 in 10,000 bone marrow cells[9], but continuously sustain blood production[10–12]. HSPCs can differentiate into common myeloid precursors and megakaryocyte-erythroid precursors (MEPs), the latter of which give rise to platelets and erythrocytes[13]. Extensive research has mostly been directed towards understanding how the endothelial cells at the first hematopoietic sites give rise to HSPCs[4,5,14]. However, increasing evidence has pointed to the capacity of endothelial cells in tissues outside of the AGM, such as the lung[15,16] and the head[17], to undergo EHT.

In hematopoietic niches, endothelial cells also play crucial roles in regulating HSPC maintenance and pluripotency[18,19]. Endothelial cells in the intermediary niches (the fetal liver in mammals and the caudal hematopoietic tissue (CHT) in zebrafish) and the bone marrow secrete angiocrine factors that sustain HSPC quiescence and self-renewal[20]. Endothelial-specific deletion of factors required for survival of endothelial cells or maintenance of HSPCs leads to loss of HSPCs[18,20,21], highlighting the importance of endothelial-HSPC communications in the niche. Whether endothelial cells can support the maintenance of HSPCs in organs other than the well-studied intermediary niches[14,22] and the bone marrow remains poorly understood. This knowledge gap limits our understanding of how differences in the locations where hematopoietic cells reside contribute to heterogeneity in HSPC characteristics[23,24].

One subpopulation of endothelial cells proposed to undergo EHT is the endocardial cells (EdCs), which form the endothelial lining of the heart. The endocardium and the contractile myocardium, which consists of cardiomyocytes (CMs), compose the first layers of the developing heart[25,26]. EdCs possess a distinct molecular profile compared with other endothelial cells[27–29] and a degree of plasticity to differentiate into other cells[30], including valve mesenchymal cells and coronary vessels[31–34]. There are also surprising findings regarding the contribution of EdCs to CM[35,36] or mural cell[37] populations.

Research concerning endocardial contribution to the hematopoietic lineage is contentious. In mouse, several studies have reported a subset of early-stage EdCs differentiating into hematopoietic cells of erythroid, myeloid, and macrophage identities[38–42]. The function of EdCs in giving rise to macrophages has been proposed to be particularly relevant, with cardiac valves being hyperplastic in the absence of endocardial-derived macrophages[39]. These studies also reported higher phagocytic activities in endocardial-derived macrophages compared with other macrophages[30], suggesting that cardiac-derived macrophages are functionally distinct from others[39]. However, other recent studies that developed endocardial-specific Cre lines performed careful lineage tracing of EdCs during development and reported an almost complete absence of endocardial-derived macrophages or circulating hematopoietic cells in mouse embryos[43,44]. Thus, the prevalence of endocardial-to-hematopoietic transition in mouse remains elusive. Reliance on using fixed and sectioned cardiac tissues for imaging, coupled with potential challenges with the specificity of Cre expression in the existing transgenic lines, have also raised difficult obstacles in addressing this intriguing question in mouse.

In this work, we use zebrafish as a model to longitudinally image the heart at high resolution and manipulate cardiac and hematopoietic development. The cardiovascular and hematopoietic systems share many similarities between zebrafish and mammals[45,46], with endothelial cells in the dorsal aorta (DA; first site of hematopoiesis) undergoing EHT[47], migrating into the CHT (intermediary niche)[48], and finally into the kidney (presumptive adult niche)[48]. A recent study uncovered a potential for EdCs to undergo EHT prior to 24 hpf, with most endocardial-derived cells fated to become neutrophils[49]. In our study, combining high-resolution imaging in live animals and single-cell transcriptomic analysis, we provide several lines of evidence that the endocardium can not only give rise to hematopoietic cells but also maintain a stable population of hematopoietic cells, which consists mainly of HSPCs/MEPs.

## Results

### A subset of endocardial cells expresses enriched levels of hematopoietic genes

The endocardium has emerged as a dynamic tissue showing remarkable plasticity[30]. Most studies on endocardial morphogenesis focus on specification and patterning of the valve[50–53], but the involvement of non-valve EdCs in driving endocardial tissue morphogenesis remains largely unexplored. To uncover other endocardial subpopulations, we performed single-cell RNA sequencing from EdCs isolated by fluorescence-activated cell sorting (FACS) of Tg(kdrl:nls-mCherry)-positive cells from dissected hearts (Fig. S1A). We focused on two developmental time points, 48 and 72 hpf, to identify morphogenetic processes that pattern the endocardium as the heart continues to loop and grow, valves take shape, and trabecular networks start to form[54].

From this dataset, we classified 11 clusters of EdCs with distinct enriched gene signatures (Fig. 1A, S1B, Supplementary Table 1). We identified two clusters that exhibit enrichment of genes involved in valve development[53,55,56], with one cluster composed of 1830 cells being consistent with valve EdCs (alcama, has2, hey2) and the other composed of 359 cells consistent with fibroblast-like valve interstitial cells (col1 genes, postnb, twist1b) (Fig. 1A, S1B, Supplementary Table 1). The largest cluster is composed mostly of 2274 non-valve quiescent EdCs in the G1 phase (Fig. 1A, S1D). Two other clusters are primarily composed of non-valve EdCs that were undergoing DNA synthesis (620 cells in the S phase) or dividing (657 cells in the G2/M phase) (Fig. 1A, S1D). A small cluster of 299 cells expresses genes consistent with lymphatic endothelial cells (lyve1a/b, flt4, mafbb; Fig. 1A, S1B, Supplementary Table 1). The remaining clusters were not as clearly defined in their transcriptomic signatures, suggesting that they are previously unidentified endocardial subpopulations with currently unknown functions.

Interestingly, one cluster most distinct from the other clusters and found almost exclusively at 72 hpf (14 cells at 48 hpf and 376 cells at 72 hpf) exhibits an enrichment of genes that promote hematopoiesis or are expressed in differentiated hematopoietic cells (Fig. 1A, B; "endocardial-hematopoietic cluster" or EHC). At 72 hpf, the EHC accounts for 9.1% of total EdCs (376 out of 4,125 cells). Within the EHC, the majority of cells express high levels of genes known to be involved in promoting HSPC fate in the CHT[57], including myb, drl, drll.2, hdr, ikzf1, and gata2b (Fig. 1B, C, S1C, S2A–E). Many EHC cells also express genes involved in erythrocyte differentiation, including gata1a and hbae3 (Fig. 1C, S1C, S2G). Two additional subclusters express high levels of platelet-promoting genes[58], including mpl and nfe2 (Fig. 1B, S2I, 165 cells). Together, the cells expressing genes enriched in HSPCs, erythrocytes, and platelets constitute 360 out of the 390 cells (Fig. 1C, S2A–E, G, I), suggesting that most cells within the EHC are HSPC/MEPs. Although it has recently been reported that hematopoietic cells derived from the endocardium at 24 hpf are primarily neutrophils[49], we found only a small subset of cells (15 cells) among 72 hpf EdCs expressing high levels of neutrophil markers (mpx and lyz) within subcluster 2 (Fig. 1C, S2A, F). Another small subcluster of 24 cells expresses macrophage-enriched markers (mpeg1.1, mpeg1.2, and irf8) (Fig. 1C, S2A, H). However, we could not detect any cell differentiation trajectories that uncover populations of EdCs in the process of EHT, likely due to low numbers of EdCs actively undergoing EHT at the timepoint of cell isolation. In conclusion, our single-cell transcriptomic dataset uncovers a previously unknown subgroup of EdCs with a hematopoietic signature.

### Hematopoietic cells are stably attached to the endocardium

We sought to visualize the presence of the hematopoietic cells within the endocardium by performing confocal imaging of transgenic lines that label HSPCs (Tg(cd41:GFP) and Tg(Runx1:mCherry) expression), as well as endothelial cells (Tg(kdrl:GFP) and Tg(kdrl:nls-mCherry) expression) at 50, 74, 98, and 120 hpf. At 50 hpf, we observed almost

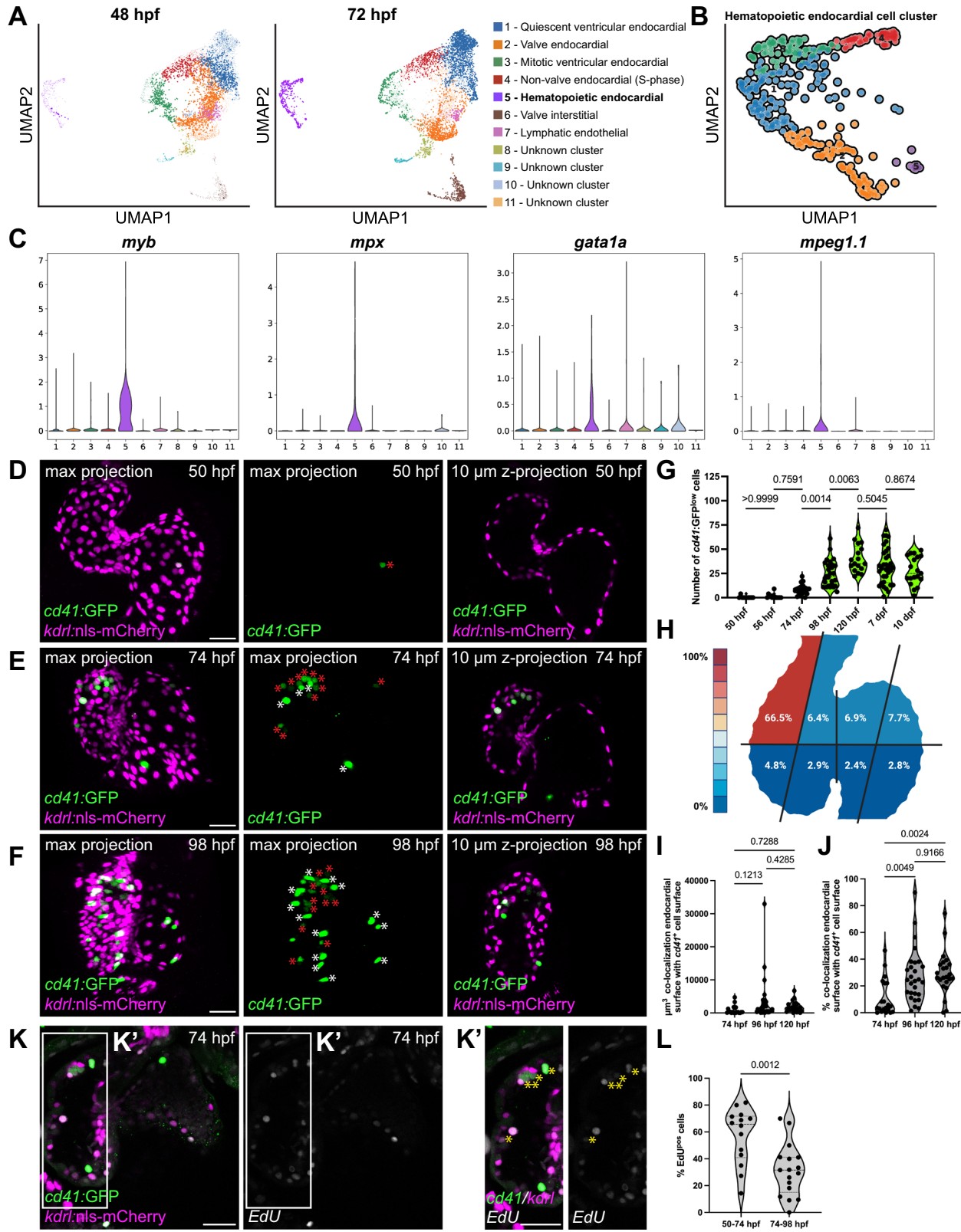

no hematopoietic cells attached to the endocardium (Fig. 1D), consistent with our RNA-sequencing dataset that shows almost no hematopoietic cells at 48 hpf (Fig. 1A). Strikingly, at 74 hpf, we observed hematopoietic cells, the majority of which were positive for the HSPC/platelet marker (*Tg(cd41:*GFP)⁺ cells; Fig. 1E, F; Supplementary Movies 1, 3, 4), attached to the endocardium. Many hematopoietic cells were also positive for the erythrocyte marker (*Tg(gata1:*DsRed)⁺; Fig. S3A),

but only very few were positive for the neutrophil marker (*Tg(mpx:*GFP)⁺; Fig. S3B), consistent with our single-cell sequencing results (Fig. 1A). Almost all the hematopoietic cells attached to the endocardium exhibited a rounded shape and nucleus, suggestive of undifferentiated HSPCs, instead of mature erythrocyte or myeloid morphologies that are larger and elongated (Fig. 1E, F). Importantly, these hematopoietic cells were not part of the circulating blood as

**Fig. 1 | Hematopoietic cells are stably maintained and grow in numbers on the endocardium. A** Single-cell RNA sequencing analysis of EdCs sorted by FACS for *Tg(kdrl:*nls-mCherry) expression at 48 and 72 hpf. The UMAPs reveal a subcluster of 390 cells that expresses genes affiliated with hematopoietic differentiation mostly at 72 hpf. **B** UMAP of the hematopoietic endocardial cell cluster indicates 5 sub-clusters. **C** Violin plots showing that genes enriched in hematopoietic cells at 72 hpf, such as *myb* (HSPCs/MEPs), *mpx* (neutrophils), *gata1* (erythrocytes), and *mpeg1.1* (macrophages), are enriched in this endocardial hematopoietic cluster, with HSPCs/MEPs and erythrocytes being the predominant cell types.
**D–F** Representative confocal images of *Tg(cd41:*GFP) expression labeling HSPCs (low GFP; red asterisks) and platelets (high GFP; white asterisks) attached to the endocardium marked by *Tg(kdrl:*nls-mCherry) expression. **D** At 50 hpf, *cd41:*GFP$^+$ cells are almost absent in the heart. **E, F** From 74 hpf onwards, *cd41:*GFP$^+$ cells are attached to the endocardium. **G** HSPC (*cd41:*GFP$^{low+}$) numbers progressively increase from 56 hpf and remain until at least 10 dpf. $n = 10$ (50 hpf), 13 (56 hpf), 19

(74 hpf), 27 (98 hpf), 17 (120 hpf), 33 (7 dpf), and 19 (10 dpf) hearts. **H** Schematic heat map representing the positional mapping of *cd41:*GFP$^+$ cells, the majority of which (66.5%) attaches to the outer curvature of the ventricle. $n = 20$ hearts. **(I, J)** Quantification of co-surface localization of *cd41:*GFP$^+$ cells and endocardium labeled by *Tg(kdrl:*BFP-CAAX) expression in the ventricle. HSPCs share 11%, 27.4%, and 29.3% of surface areas with the ventricular endocardium at 74, 96, and 120 hpf, respectively. $n = 20$ (74 hpf), 28 (96 hpf), and 23 (120 hpf) hearts, respectively. **K, K′** Representative confocal images of EdU pulse labeling assays (50–74 hpf) in *Tg(cd41:*GFP), *Tg(kdrl:*nls-mCherry) larvae. Most *cd41:*GFP$^+$ cells attached to the heart are EdU-positive at 74 hpf (yellow asterisks). **L** Quantification of the proportions of *cd41:*GFP$^+$ cells positive for EdU after a pulse assay between 50–74 and 74–98 hpf. $n = 14$ (50–74 hpf) and 17 (74–96 hpf) hearts. One-way ANOVA with Tukey's multiple comparison test (**G, I, J**) or Unpaired, two-tailed *t* test (**L**) was used. Scale bars, 30 μm (**D–F, K**). Source data are provided as a Source Data file.

observed through the imaging of beating hearts, which revealed their stable attachment to the endocardium without any movement in the cardiac lumen (Supplementary Movies 1, 2). These hematopoietic cells were never attached to the cardiac valve (Fig. 1E, F), suggesting that valve cells lack the adhesive properties for hematopoietic cells.

As most endocardial-associated hematopoietic cells appeared positive for HSPC/MEP identity (Fig. 1A–C, S1C, S2A–I; Supplementary Movies 1, 3, 4), we focused on the emergence and maintenance of these cells in the heart. We labeled them in the heart over several developmental time points using *Tg(cd41:*GFP) expression (hereafter referred to as *cd41:*GFP$^+$ cells), a commonly used and reliable marker where low GFP signal is detected in HSPCs and high GFP signal in platelets. *cd41:*GFP$^+$ cells were almost absent at 50 hpf, but started appearing in very low numbers at 56 hpf in 52% (11 out of 21) embryos (Fig. 1D, G; Supplementary Table 2). The numbers of *cd41:*GFP$^{low+}$ and total *cd41:*GFP$^+$ cells significantly increased at 74 hpf (Fig. 1E, G; average of 10.1 *cd41:*GFP$^{low+}$ cells and 23.1 *cd41:*GFP$^+$ cells per heart) and continued to rise at 98 hpf (Fig. 1F, G; average of 25 *cd41:*GFP$^{low+}$ cells and 33 *cd41:*GFP$^+$ cells per heart), and 120 hpf (Fig. 1G; average of 40.2 *cd41:*GFP$^{low+}$ cells per heart). The *cd41:*GFP$^{low+}$ cells persisted at stable numbers until at least 10 days post fertilization (dpf) (Fig. 1G; 32.7 and 27.8 *cd41:*GFP$^{low+}$ cells at 7 and 10 dpf, respectively). By 74 hpf, 96% of the hearts had *cd41:*GFP$^+$ cells on their endocardium; from 100 hpf onwards, all hearts contained *cd41:*GFP$^+$ cells (Supplementary Table 2). Positional mapping of *cd41:*GFP$^{low+}$ cells revealed the highest frequency of HSPC attachment on the luminal surface of the outer curvature of the ventricle, close to the outflow canal (Fig. 1H; Supplementary Movies 1–4; 66.5% of endocardial-attached HSPCs), uncovering a regional preference for hematopoietic attachment.

To measure the adherence of the HSPCs to the endocardial surface, we quantified the degree of overlap between the HSPCs, as labeled by *Tg(cd41:*GFP) expression, and the endocardium, as labeled by *Tg(kdrl:*BFP-CAAX) expression. We found that over time, there was a progressive increase of the surface of HSPCs that overlapped with the endocardium, indicating that HSPCs gain significantly higher contact area with the endocardium, particularly between 74 and 96 hpf (Fig. 1I, J, S4A–C). The increasing contact areas between these two cell types suggest that the HSPCs become more adherent to the endocardium over time.

To strengthen our findings that the endocardial hematopoietic cells are HSPCs, we imaged and quantified *Tg(Runx1:*mCherry)-positive cells. *Runx1* expression has been consistently used not only in zebrafish[22], but also in mouse to mark HSPCs and hemogenic endothelial cells[59]. Similar to *cd41:*GFP$^+$ cells, we uncovered a progressive increase in the number of *Runx1:*mCherry$^+$ hematopoietic cells attached to the endocardium starting from 56 hpf (Fig. S5A, B). Imaging of *Tg(cd41:*GFP) and *Tg(Runx1:*mCherry) expression in the same animal revealed that most cardiac-residing hematopoietic cells were positive for both low GFP and mCherry expression (Fig. S5C, D),

consistent with previous co-expression studies of these HSPC-marking transgenes[22]. We also performed fluorescence in situ hybridization for *myb*[60], another HSPC marker that was also enriched in the EHC of our single-cell sequencing dataset (Fig. 1C, S2D), and observed its expression in the EdCs in the outer curvature of the ventricle (Fig. S5E). Based on their gene expression profiles and cell morphology, we conclude that HSPCs/MEPs are stably maintained on the endocardium.

We then assessed whether the heart supports HSPC proliferation, which is an essential function of a hematopoietic niche, by performing an EdU pulse labeling assay. Between 50 and 74 hpf, when we observed the highest growth of endocardial-residing hematopoietic cells, the majority (57%) of *cd41:*GFP$^+$ cells associated with the heart were EdU-positive (Fig. 1K, L). Between 74 and 98 hpf, the proportion decreased to 31% of *cd41:*GFP$^+$ cells (Fig. 1L). These results suggest that hematopoietic cells can proliferate on the luminal surface of the endocardium.

## Some endocardial cells can undergo EHT

The appearance of endocardial-residing hematopoietic cells raised the question as to whether these cells arise de novo from EdCs or migrate from the DA. A significant limitation we encountered was a lack of EdC- or DA-specific transgenic lines. Thus, we used a photoconversion-based approach for cell lineage tracing. We utilized the Kaede protein that turns from green to red upon excitation with UV wavelength[53] and when expressed specifically in endothelial cells ((*Tg(fli1a:*Gal4; *Tg(UAS:*Kaede), hereafter referred to as *Tg(fli:*Kaede)), enables us to track a spatially confined group of endothelial cells.

We first photoconverted the entire endocardium (Fig. 2A, A′) at 22 hpf, when the heart is not yet fully contractile and the atrium has not yet connected to the cardinal vein. We grew the photoconverted animals for two days and imaged the hearts at 74 hpf, the stage when we observed many endocardial-residing hematopoietic cells. We indeed found photoconverted Kaede-red cells that appear to be budding off from the endocardium at 74 (Fig. 2B, B′), 96 (Fig. S6A), and 120 (Fig. S6B) hpf, suggesting that EdCs give rise to hematopoietic cells that persist in the heart. Interestingly, when we examined other hematopoietic niches, we found photoconverted Kaede-red cells in the CHT (Fig. 2C), the intermediary niche that allows for HSPC population expansion, as well as the thymus (Fig. 2D), where adaptive immune cells reside and proliferate. Based on the Kaede-red/Kaede-green color ratio, we quantified photoconverted cells that reside in non-photoconverted tissues. We observed approximately 16.7 and 0.92 endocardial-derived cells residing in the CHT (Fig. 2E) and the thymus (Fig. 2F), respectively. Endocardial-derived Kaede-red cells were found in all the CHTs we imaged, but only in 57% (8 out of 14) of the thymuses we imaged. The low numbers of endocardial-derived cells in the thymus compared with the CHT suggest that most endocardial-derived hematopoietic cells are not fated to become immune cells. Our data suggest that the endocardium gives rise to cells that not only remain

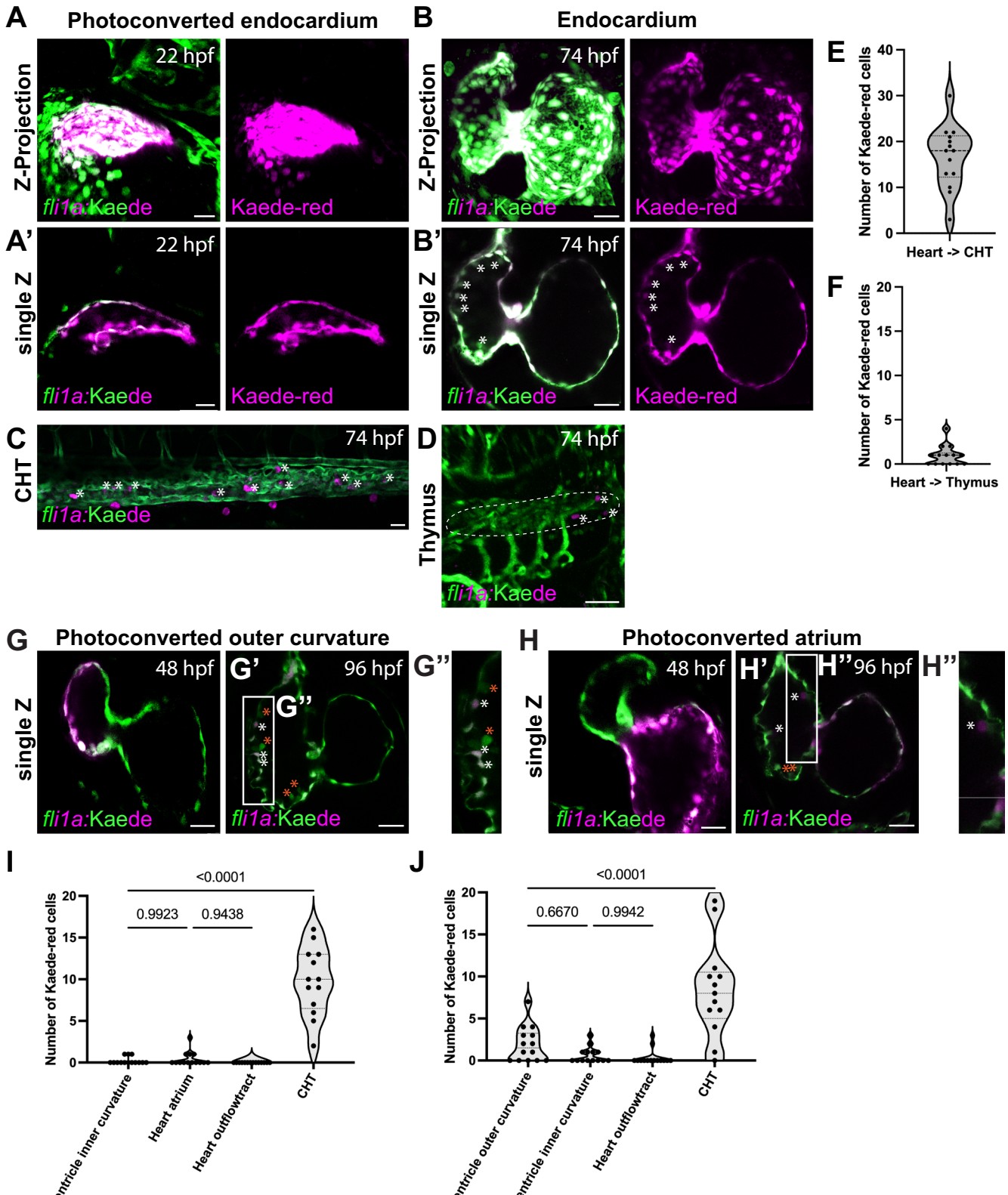

attached to the endocardium, but also circulate and eventually reside in extracardiac hematopoietic niches.

As our analysis showed that most hematopoietic cells were found in the outer curvature of the cardiac ventricle (Fig. 1H), we investigated whether EdCs possess heterogeneous, region-specific capability to give rise to hematopoietic cells. We photoconverted different regions of 48 hpf hearts, which exhibit spatially distinct regions: the atrium, the ventricular outer curvature (VOC), and ventricular inner curvature (VIC). We then grew the animals for 24 or 48 h and quantified Kaede-red cells in the other regions of the heart as well as in the CHT at 72 and 96 hpf. Interestingly, we observed Kaede-red cells in the VOC not only when cells from the same region were photoconverted (Fig. 2G–G", S7A, white asterisks), but also when atrial EdCs were photoconverted (Fig. 2H–H", S7E, white asterisks; average of 1.9 cells). Whereas VOC-

**Fig. 2 | Endocardial cells contribute to cardiac-residing hematopoietic cells.**
**A–B'** Confocal Z-stack projection (**A**) and single confocal plane (**A'**) of photo-converted endocardium at 22 hpf, and follow-up imaging of the same animals (**B**, confocal Z-stack projection; **B'**, single confocal plane) 52 h post-conversion. Extruding Kaede-red[+] cells are observed on the endocardium at 74 hpf (white asterisk). **C, D** Confocal image of 74 hpf *fli*:Kaede[+] caudal hematopoietic tissue (CHT; **C**) and thymus (**D**) from larvae with their endocardium photoconverted at 22 hpf. Kaede-red[+] cells (white asterisk) surrounded by endothelial cells are visible in the CHT (**C**); few Kaede-red[+] cells visible in the thymus (white asterisks) (**D**). **E, F** Quantification of Kaede-red[+] cells from the photoconverted endocardium (22 hpf) reveals 16.7 and 0.92 endocardial-derived cells in the CHT (74 hpf, **E**) and thymus (74 hpf, **F**), respectively, 52 h post-conversion. *n* = 14 CHT and 14 thymi. **G–H'** Single confocal plane of a heart with photoconverted endocardium in the ventricular outer curvature (**G, G'**) or the atrium (**H, H'**) at 48 hpf and at 48 h post-conversion (**G'–H'**). **G", H"** Closeup single plane of the 96 hpf *fli*:Kaede[+] heart,

where EdCs in the ventricular outer curvature (**G"**) or the atrium (**H"**) of the same animals had been photoconverted at 48 hpf. **G"** Kaede-red[+] cells observed on the ventricular endocardium (white asterisks) near Kaede-green[+] cells (red asterisks). **H"** Kaede-red[+] cells derived from the atrium observed close to the atrial and ventricular endocardium (white asterisks). Kaede-green[+] cells (red asterisks) are also present in the ventricle. **I, J** Quantification of Kaede-red[+] cells derived from the ventricular outer curvature (**I**) or the atrium (**J**) found in other regions of the heart and the CHT, 48 h post-conversion. Several Kaede-red[+] cells from the ventricular outer curvature found in the CHT, but not in other heart regions. Atrial-derived Kaede-red[+] cells are present in the CHT and ventricular outer curvature. *n* = 14 hearts with photoconverted ventricular outer curvature and 14 hearts with photoconverted atrium, respectively. One-way ANOVA with Tukey's multiple comparison test (**I, J**) was used. Scale bars, 30 μm (**A–D, G–H"**). Source data are provided as a Source Data file.

derived cells were not frequently found in other cardiac regions (Fig. 2I; 6 out of 14 (43%) hearts at 96 hpf), atrial-derived Kaede-red cells were attached to the VOC (Fig. 2H', H", J; 9 out 14 (64%) hearts at 96 hpf). Quantification of endocardial-derived cells revealed that the VOC and the atrium contribute to approximately the same numbers of cells in the CHT (Fig. 2I, J, S7B–D, S7F–H; 3.4 and 2.2 cells at 72 hpf, 9.1 and 7.8 cells at 96 hpf from the VOC and atrium, respectively). The VIC contributed almost no cells to the other regions of the heart (Fig. S8A–C, E), and fewer cells than the other cardiac regions to the CHT particularly at 72 hpf (Fig. S8D, E; average of 0.5 cells at 72 hpf and 6.6 cells at 96 hpf). Thus, our data suggest that the endocardium in the VOC and atrium have the highest propensity to differentiate into hematopoietic cells that remain in the heart or integrate into other niches.

To further test the potential of EdCs to autonomously generate hematopoietic cells in isolation, we dissected and cultured the hearts ex vivo for 24 h. We adopted an established protocol for culturing embryonic hearts[61], which were extracted from 48 hpf embryos positive for *Tg(cd41*:GFP) and/or *Tg(Runx1*:mCherry*)* expression (Fig. 3A). At this stage, the heart consists of only the endocardium and myocardium[45]. Imaging the hearts immediately post dissection showed that only a minority of them contained hematopoietic cells at 48 hpf (Fig. 3B, C; 2 out of 13 hearts were *cd41*:GFP[+] (15%); 3 out of 12 hearts were *Runx1*:mCherry[+] (25%)). We grew the 48 hpf hearts in Dulbecco's media supplemented with fetal bovine serum for 24 h, and selected the ones that were still contracting in the media for further analysis. Strikingly, the majority of the hearts grown ex vivo for 24 h contained *cd41*:GFP[+] (16 out of 17 hearts (94%)) or *Runx1*:mCherry[+] (15 out of 19 hearts (79%)) cells (Fig. 3B, C). These results further indicate that between 48 and 72 hpf, EdCs can give rise to HSPCs without contribution from the DA or other extra-cardiac sources.

Furthermore, using a custom-built light sheet microscope, we recorded high-speed movies of the beating heart (Fig. 3D, Supplementary Movie 5). We followed individual EdCs over a 7-hour period starting from 72 hpf in the beating hearts of *Tg(cd41*:GFP)[+], *Tg(kdrl*:nls-mCherry)[+] larvae (Fig. 3E–J, Supplementary Movie 5). From the reconstructed images, we identified EdCs that did not initially express *Tg(cd41*:GFP) at the start of imaging but then gradually turned on *Tg(cd41*:GFP) expression (Fig. 3E–J, yellow asterisk in Supplementary Movie 5). These EdCs were found in several regions of the cardiac ventricle and remained integrated within the endocardium by the end of the imaging period. Together, our results suggest that some EdCs can undergo EHT, activate HSPC/MEP markers, and contribute to hematopoietic populations in the heart and other niches.

### Dorsal aorta endothelial cells are a major contributor to endocardial-residing hematopoietic population

We next investigated the contribution of endothelial cells from the DA to cardiac-residing cells by photoconverting them and imaging the hearts 48 h post-conversion. We narrowed down specific DA regions

that might contribute cells to the heart and converted the anterior-most DA region (Fig. 4A), the middle DA region above the yolk extension (Fig. 4D), and the posterior-most DA region (Fig. 4G). Interestingly, we found that many cells derived from the DA end up residing in the endocardium. While the yolk extension DA region gave rise to some cells (Fig. 4E, F, M; average of 2.25 cells at 74 hpf and 4.5 cells at 98 hpf), the posterior DA region yielded the most cardiac-residing cells (Fig. 4H, I, M; average of 6 cells at 74 hpf and 8 cells at 98 hpf). In contrast, the anterior DA region contributed almost no cells to the heart (Fig. 4B, C, M; average of <1 cell at 74 and 98 hpf). Photoconverting the middle and posterior DA regions at 22 hpf (Fig. 4J) yielded an average of 9.75 DA-derived cells attached to the endocardium at 74 hpf (Fig. 4K, M), which amounted to 42% of the total *cd41*:GFP[+] cells in 74 hpf hearts (average of 23.1 cells). At 98 hpf, the number of DA-derived Kaede-red cells attached to the endocardium significantly increased to 21 cells (Fig. 4L, M), which represented 63% of the total *cd41*:GFP[+] cells (average of 33 cells). These results indicate that DA-derived cells attach and significantly expand in numbers once attached to the endocardium, which could be driven by proliferation or capture of more circulating cells on the endocardium.

We also found that photoconverting the DA at 38 hpf still led to the appearance of photoconverted cells in the heart (Fig. 5G, H), indicating that DA endothelial cells retain their capacity to undergo EHT at these later stages. In 74 hpf hearts, we found similar numbers of Kaede-red cells derived from the DA endothelial cells photoconverted at 38 hpf compared with ones photoconverted at 22 hpf (Fig. 5M; 7.8 cells from 38-hpf DA, 9.75 cells from 22 hpf DA), suggesting that most or all DA endothelial cells retain their EHT capacity and attachment capability to the endocardium at 38 hpf. Altogether, these results show that many cardiac-residing hematopoietic cells derive from the DA, particularly the posterior-most region of the DA.

### Hematopoietic cells attach to the endocardium via an Integrin α4-Vcam1-dependent adhesion mechanism

We hypothesized that an active mechanism in which adhesion molecules mediate endocardial-hematopoietic attachment was necessary to withstand the high-flow intracardiac environment. One essential ligand-receptor pair that mediates HSPC homing into their niche is the ligand Vcam1 (Vascular cell adhesion molecule 1), which is mainly present on the endothelial cell surface of HSPC niches, and its receptor Itgα4 (Integrin α4), which is present on the surface of HSPCs[62]. In zebrafish, loss of either *itga4* or *vcam1b* does not impact EHT in the DA but does result in the failure of HSPCs to seed the CHT[63].

Our single-cell transcriptome dataset showed that *itga4* expression was present at high levels in the EHC (Fig. 5A; Cluster 5). *vcam1b* expression is high in other EdCs that might be hematopoietic niche cells (Fig. 5A; Cluster 1-4, 8) but not in the EHC itself (Fig. 5A; Cluster 5) or in the valve cell population (Fig. 5A; Cluster 6). Using an RNAscope assay to detect *vcam1b* mRNA in the heart, we found that within the

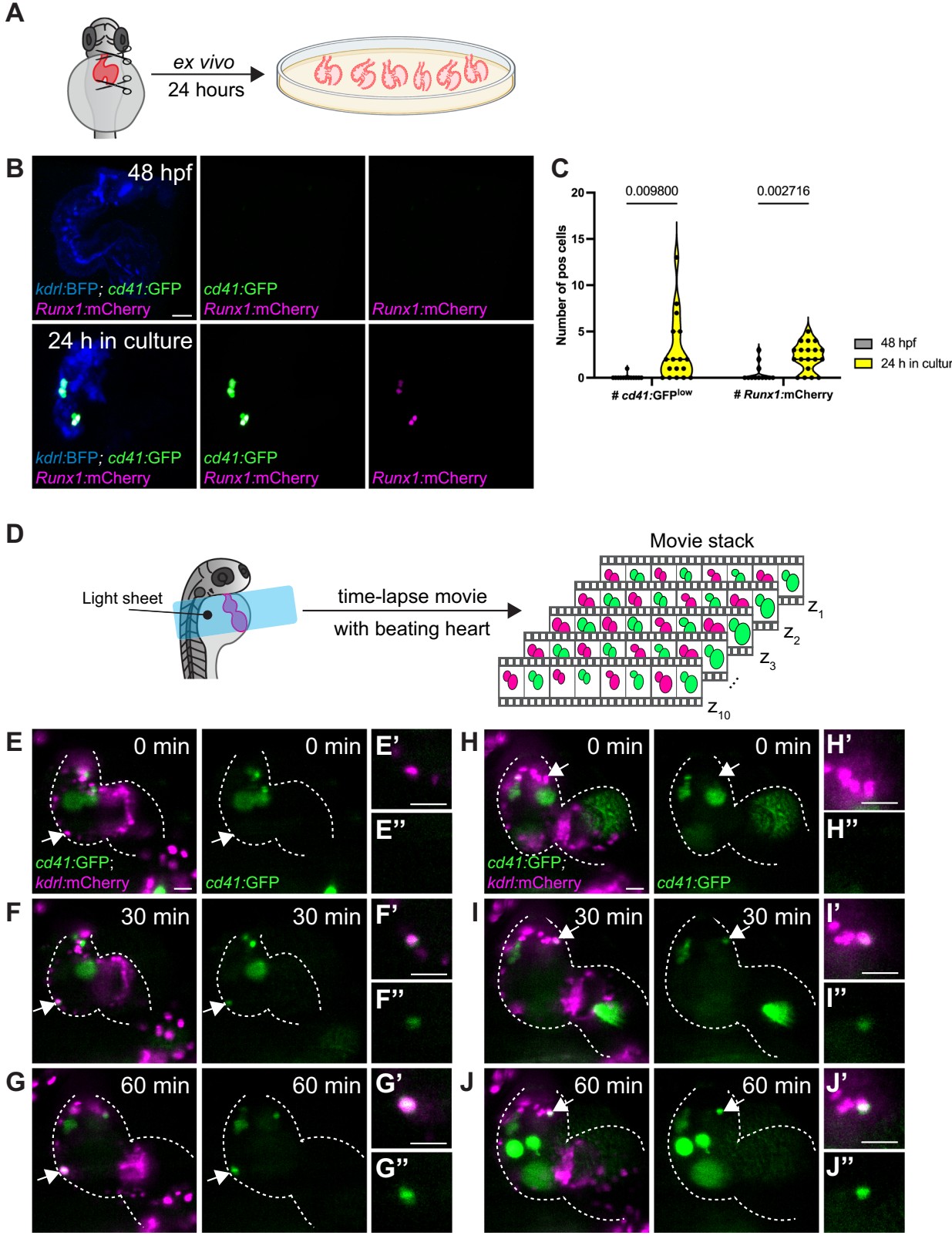

endocardium, *vcam1b* appeared enriched in the EdCs in the VOC (Fig. 5B, B'). In contrast, *vcam1b* did not appear detectable in the valve EdCs (Fig. 5B, B"). The high *vcam1b* expression in ventricular EdCs suggests that the endocardium exhibits similarities in molecular signatures with endothelial cells in HSPC niches and might explain the preferential adhesion of hematopoietic cells to EdCs in the VOC but not in the valve.

We tested the function of Vcam1 and Itgα4 in HSPC-endocardial attachment by quantifying *Runx1*:mCherry⁺ cells in *itga4* and *vcam1b* mutants or *cd41*:GFP⁺ cells in animals with morpholino-mediated knockdown of *itga4* or *vcam1b*. *itga4* and *vcam1b* mutants and morphants do not appear severely defective in development, with wild-type like body lengths and cardiac morphology (Fig. S9A, B), as well as unaffected cardiac valve function (Supplementary Movies 6-8).

**Fig. 3 | Ex vivo transplants and longitudinal light sheet imaging of the beating heart show endocardial cells activating HSPC markers. A** Schematics of ex vivo growing of hearts extracted at 48 hpf and cultured for 24 h. **B** Representative confocal images of a 48 hpf extracted *Tg(Runx1:mCherry), Tg(cd41:GFP), Tg(kdrl:BFP-CAAX)* heart, showing no *Runx1*:mCherry⁺ or *cd41*:GFP⁺ cells. After culturing for 24 h, *Runx1*:mCherry⁺ and *cd41*:GFP⁺ cells are present in the heart. **C** Significantly higher numbers of *Runx1*:mCherry⁺ and *cd41*:GFP⁺ cells are observed in hearts at the end of the 24 h culture compared with 48 hpf extracted hearts. $n = 13, 17, 12$, and 19 hearts, from left to right. **D** Schematics of a 72 hpf zebrafish larva at the illumination axes of the light sheet microscope. Movie stack

acquisitions (one movie per plane) were used to reconstruct a synchronization of the beating heart every 30 min for 7 h. **E–J** Single plane of an endocardial cell in the ventricular outer curvature (**E–G**, magenta, white arrow) or in the ventricular inner curvature (**H–J**, magenta, white arrow). At 0 min (**E, H**), no *Tg(cd41*:GFP) expression is observed. At 30 min (**F, I**), *Tg(cd41*:GFP) expression appears noticeable (**F**); at 60 min (**G, J**), strong *Tg(cd41*:GFP) expression is visible in endocardial cells that remain integrated in the heart. The hearts are outlined in dashed lines. **C** Unpaired, two-tailed *t* test was used. Scale bars, 30 μm (**B, E–J**). Source data are provided as a Source Data file. Figure A, D created with BioRender.com released under a Creative Commons Attribution-NonCommercial-NoDerivs 4.0 International license.

---

Quantification of *Runx1*:mCherry⁺ cells in *itga4* and *vcam1b* mutants compared with heterozygous siblings revealed an almost complete loss of *Runx1*:mCherry⁺ cells (Fig. 5C, D; asterisks indicate *Runx1*:mCherry⁺ cells in Fig. 5C). Similarly, significantly lower numbers of *cd41*:GFP⁺ cells were attached to the endocardium of *itga4* and *vcam1b* morphants at 74 hpf (Fig. 5E, F). These results indicate that Itgα4-Vcam1b-based adhesion is necessary to maintain endocardial hematopoietic cells, and that the endocardium may share some niche characteristics with the CHT.

We then determined whether the endocardium fails to mediate attachment of circulating hematopoietic cells from the DA upon loss of *itga4* or *vcam1b* function, similar to what was reported for the CHT vasculature. We photoconverted the DA endothelial cells in control, *itga4*, and *vcam1b* morphants at 38 hpf (Fig. 5G, I, K), and quantified endocardial-residing hematopoietic cells at 74 hpf (Fig. 5H, J, L). We found a substantial reduction of DA-derived hematopoietic cells on the endocardium when *itga4* or *vcam1b* was knocked down (Fig. 5G–M; averages of 2.5 and 3.5 cells in *itga4* and *vcam1b* morphants, respectively, compared with 7.8 cells in control). These results indicate that Itga4 and Vcam1b promote the attachment of circulating hematopoietic cells to the endocardium.

### Hematopoietic cell attachment to the endocardium is at least partly independent of cardiomyocyte trabeculation and reduced shear stress

Our results narrowed the time window when endocardial-residing hematopoietic cells appear in the heart to be between 56 and 72 hpf. During this period, trabecular cardiomyocytes delaminate and form ridges in the cardiac lumen[64]. These ridges are computationally predicted to produce pockets of low shear stress[65], which led us to hypothesize that they allow the hematopoietic cells to attach to the endocardium. To test this hypothesis, we used two hypotrabeculation models: *nrg2a* mutants, which fail to activate ErbB2[66], and the ErBB2 pharmacological inhibitor PD168393 (PD)[67]. In both models, the endocardium and myocardium remained single-layered without trabecular ridges (Fig. 6A–C). However, at 74 hpf, *cd41*:GFP⁺ cells remained attached to the endocardium in both hypotrabeculation models (Fig. 6A–C). Quantification of *cd41*:GFP⁺ cells shows no significant reduction in the number of endocardial-attached HSPCs (Fig. 6D, E), and no changes in HSPC position within the heart of *nrg2a* mutants (Fig. S10A, B), indicating that cardiac trabeculation is not necessary for hematopoietic cells to attach to the endocardium.

Although the endocardium is similar to the CHT in its dependence on Vcam1-Itgα4 for HSPC lodgment, it exhibits apparent differences in tissue organization. Unlike the fenestrated CHT vasculature[22], the endocardium is constantly exposed to blood flow. Combined with our observations that cardiac hypotrabeculation did not affect HSPC/MEP numbers in the heart, we hypothesized that hematopoietic cell attachment to the endocardium was at least partly independent of mechanical forces. Stopping the heartbeat in *tnnt2a* loss-of-function morphants, or with high concentrations of the anesthetic Blebbistatin, led to failure of the heart to balloon and grow, which unsurprisingly led to the almost-complete absence of HSPCs in the heart (Fig. S11A, B). To

mitigate secondary defects from severe cardiac problems, we used a low concentration of anesthetics from 48 to 74 hpf (MS-222/Tricaine or Blebbistatin) or a low dose of *tnnt2a* morpholino to reduce the heartbeat by around 30% (Fig. S11C). Consistent with our hypothesis, the numbers of endocardial-residing *cd41*:GFP^low+ cells were not significantly altered upon reduced cardiac contraction and reduced shear stress (Fig. 6F–J). These data suggest that the attachment of hematopoietic cells to the endocardium is at least partly independent of biomechanical forces from the blood flow or heartbeat.

### Abrogation of primitive erythropoiesis leads to increased cardiac-residing HSPCs/MEPs

The emergence of hematopoietic cells on the endocardium after 56 hpf suggests that this process is driven by the definitive and not primitive hematopoiesis. We investigated whether abrogating primitive erythropoiesis affects HSPC/MEP appearance in the heart by knocking down *gata1*, a gene encoding a transcriptional promoter of primitive erythropoiesis[68]. When the *gata1* morpholino is injected at sub-optimal concentrations, *gata1* morphants lack red blood cells for only the first 96 h of development[65] (own observations). Surprisingly, *gata1* knockdown led to a significantly increased number of endocardial-attached *cd41*:GFP^low+ cells (Fig. 7A, B). Analysis of *gata1* morphant hearts at 48 and 54 hpf shows that *cd41*:GFP^low+ cells did not emerge or attach earlier than in control embryos (Fig. S12A-E), but instead significantly increased at 74 hpf when *cd41*:GFP^low+ cells were consistently found in almost all wild-type hearts (Fig. 7D; Fig. S12E).

Erythrocytes constitute a significant source of shear stress in the vascular lumen, which promotes HSPC formation in the zebrafish DA[69] and the mouse aorta-gonad-mesonephros[70]. Although our results suggest that HSPC/MEP emergence in the heart is partly independent of shear stress, we tested whether the increase of *cd41*:GFP⁺ cells in *gata1* morphants was due to altered biomechanical forces using a low concentration of MS-222. We still observed a trend of increasing numbers of endocardial-residing *cd41*:GFP⁺ cells in *gata1* morphants with reduced heartbeat (Fig. 7C, D), suggesting that the effects of abrogating primitive erythrocytes on HSPC/MEP presence in the heart may be independent of erythrocyte-induced biomechanical forces.

We further tested whether abrogating primitive erythrocytes induced a global response from other vascular tissues to produce or maintain an increased number of HSPCs. We quantified *cd41*:GFP⁺ cells in the CHT at 74 hpf in control (Fig. 7E) and *gata1* morphants (Fig. 7F). Interestingly, we did not observe any significant changes in the numbers of *cd41*:GFP⁺ cells (low or high GFP signal) in the CHT (Fig. 7G). These results show that abrogating primitive erythropoiesis does not lead to a non-specific global increase in HSPCs but does influence the numbers of cardiac-residing HSPCs/MEPs.

We then investigated whether the absence of primitive erythrocytes could influence the differentiation state of the increased HSPCs/MEPs in the heart. We combined *Tg(cd41*:GFP) expression to mark HSPCs and *Tg(gata1*:DsRed) expression to mark erythrocytes in the same larvae, imaged them in control and *gata1* morphants, and quantified the proportion of double-positive cells in both conditions

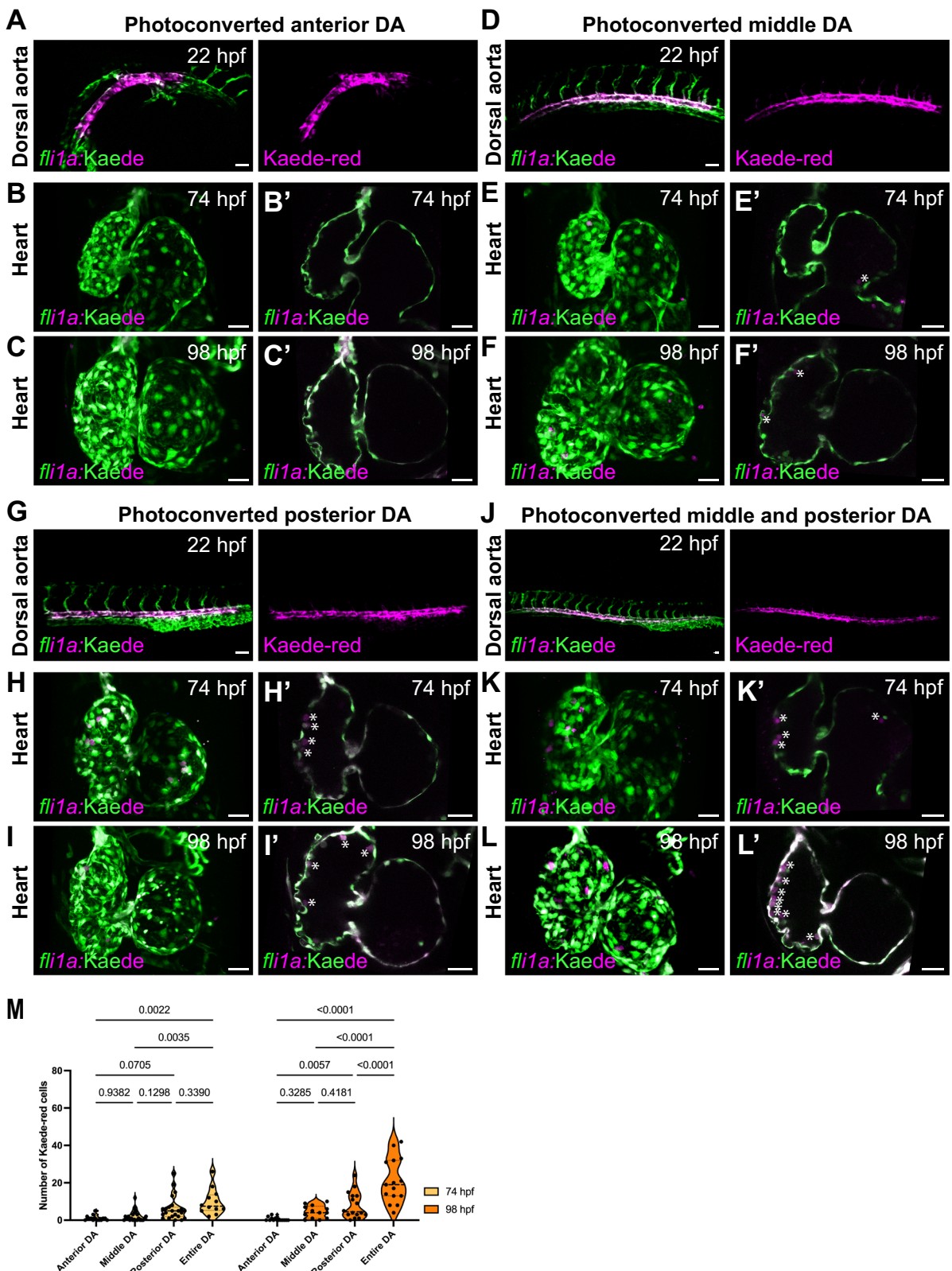

(Fig. 7H–K). We only considered *gata1*:DsRed⁺ cells that exhibited a round morphology and not the elongated mature erythrocyte morphology. In control hearts, a significant percentage (~50%; Fig. 7I) of *cd41*:GFP⁺ cells were also positive for *Tg(gata1*:DsRed) (Fig. 7H–J), suggesting that these HSPCs/MEPs are naturally primed to become erythrocytes. This proportion of *cd41*:GFP⁺; *gata1*:DsRed⁺ cells significantly increased in *gata1* morphant hearts (average of 70%;

Fig. 7H, I, K), suggesting that the lack of circulating primitive erythrocytes not only increased endocardial-associated HSPCs but also promoted their differentiation state into erythrocytes. Altogether, these data suggest that when circulating primitive erythrocytes are lost in the first few days of development, attachment of more HSPCs/MEPs primed to differentiate into erythrocytes to the endocardium was induced.

**Fig. 4 | Dorsal aorta endothelial cells are major contributors to cardiac-residing hematopoietic cells. A, D, G, J** Representative confocal images of Kaede green-to-red photoconversion of *Tg(fli1:*Kaede)⁺ endothelial cells in the anterior-most region (**A**), middle region (**D**), posterior-most region (**G**), or combined middle and posterior regions (**J**) of the dorsal aorta (DA) at 22 hpf. **B–C'** Z-stack projections and single confocal planes of the heart at 74 (**B, B'**) and 98 (**C, C'**) hpf show minimal to no contribution of anterior DA-derived Kaede-red⁺ cells to cardiac-residing cells. **E–F'** Z-Stack projections and single confocal planes of the heart at 74 (**E, E'**) and 98 (**F, F'**) hpf show moderate contribution of the middle DA region to the heart (white asterisks). **H–I'** Z-Stack projections and single confocal planes of the heart at 74 and 98 hpf demonstrate a significant contribution of posterior DA cells to the heart.

**K–L'** Photoconversion of endothelial cells in the middle and the posterior DA (combined) regions at 22 hpf lead to the highest numbers of DA-derived cells attached to the endocardium (white asterisks). **M** Quantification of DA-derived Kaede-red⁺ cells attached to the endocardium at 74 and 98 hpf. The posterior DA subregion contributes the highest numbers of cardiac-residing cells. Photoconverting endothelial cells in both the middle and posterior DA (combined) regions yields the most cardiac-residing cells. *n* = 14 (anterior), 24 (middle), 23 (posterior), and 12 (combined) hearts at 74 hpf, and 14 (anterior), 13 (middle), 16 (posterior), and 15 (combined) hearts at 96 hpf. A one-way ANOVA with Tukey's multiple comparison test was used. Scale bars, 30 μm (**A–L**). Source data are provided as a Source Data file.

## Discussion

Our results provide evidence that the zebrafish endocardium can sustain a population of HSPCs/MEPs for at least the first 10 days of development (Fig. 8). Our data are consistent with the previous report by Nakano et al.[38] that mouse EdCs in the outflow tract give rise to cells positive for the HSPC marker CD4, suggesting that a spatially confined subset of EdCs undergo EHT. However, important distinctions exist between our findings in zebrafish and the reports on mouse endocardial hematopoietic cells. First, whereas HSPCs/MEPs are only temporarily found in mouse hearts between E8.5 and E10.5, these cells persist and multiply in the zebrafish heart (Fig. 1D–J). Second, in mouse, HSPCs appear in the endocardium even earlier (E8.5) than the AGM (E10.5); we found that in zebrafish, hematopoietic cells appear later in the heart (~72 hpf) compared with the DA (~24 hpf) (Fig. 1D, E, J). Loss of Integrin α4 and Vcam1, which does not affect EHT in the DA but affects HSPC incorporation into the CHT[63], significantly reduces hematopoietic cell attachment to the endocardium, suggesting that the heart acts more as a secondary niche than a primary hematopoietic production tissue. Finally, other reports note that one of the major cells derived from EdCs is macrophages, which are particularly important for valve remodeling. We did not observe hematopoietic cell attachment to the valves, and only a minor population of macrophages in our transcriptomic dataset. Whether zebrafish EdCs generate macrophages during later development or in adults, and whether the endocardial-derived hematopoietic cells differentiate into macrophages upon valve injury remain interesting questions. In mouse, limitations with Cre lineage tracing have also called the endocardial contribution into hematopoietic lineages into question[43,44]. In contrast, our study utilizes imaging of live animals and clearly showed the persistence of the HSPCs/MEPs in the beating zebrafish heart (Supplementary Movies 1–4).

A notable limitation of our study pertains to a lack of endocardial-specific Cre line in zebrafish for definitive lineage tracing of endocardial-derived hematopoietic cells. We utilized the photo-convertible Kaede line to the best of our technical abilities and uncovered the spatiotemporal contribution of endothelial cells in the heart and the posterior DA region to cardiac-residing cells. However, we could not combine other transgenic lines labeling hematopoietic lineages with photoconverted Kaede-red⁺/Kaede-green animals because of color restrictions in fluorescence or perform immunostaining with photoconverted Kaede-red⁺ animals due to strong background fluorescence of fixed samples. Based on the single-cell sequencing data and the low contributions of endocardial-derived cells in the thymus, the resident tissue for immune cells, our results suggest that endocardial-residing/-derived HSPCs might be biased towards the erythrocyte/platelet fate. With the current limitations, we could not distinguish whether environmental signals from the endocardium result in a lineage bias of any hematopoietic cells residing in the heart, or whether hematopoietic cells derived from the endocardium are intrinsically fated to become erythrocytes or platelets. Future investigations with improved lineage tracing tools are needed to address this question, and to uncover the precise hematopoietic

identities of endocardial-derived cells. In addition, the establishment of endocardial-specific lineage tracing tools will help to investigate the capability of the endocardium to support long-term quiescence and proliferation of HSPCs throughout the zebrafish lifetime.

Although the endocardium appears to exhibit some characteristics suggestive of an HSPC niche, such as the prevalence of EdU⁺ HSPCs attached to the endocardium (Fig. 1K, L) and the high *vcam1b* expression in the EdCs that retain HSPCs (Fig. 5B–B"), it is also important to note several significant differences between known HSPC niches, particularly the CHT, and the endocardium. The endocardium is constantly under mechanical pressure from cardiac contraction and blood flow. In contrast, the endothelial cells in the CHT are fenestrated, and form pockets for the HSPCs protected against blood flow. In addition, whereas the CHT contains stromal cells that provide signaling to the HSPCs, the endocardium does not possess stromal cell populations, and is mostly in contact with the myocardium. Whether the myocardial tissue provides signaling cues to maintain the hematopoietic cells is unknown, but the myocardial trabecular network does not appear to provide structural support for the HSPCs (Fig. 6A–E). Lastly, our data indicate that, unlike the heart, the CHT does not respond to the lack of primitive erythrocytes caused by *gata1* knockdown (Fig. 7A–K), suggesting that the endocardium and the CHT play distinct functions under different physiological conditions. Thus, the endocardium provides a unique model to study how hematopoietic cells attach and proliferate under constant pressure. We are currently investigating molecular similarities and differences between the endocardium and the CHT.

Studies of EHT in the DA indicate that HSPC budding depends on blood flow-induced shear stress[69]. Similarly, multiple developmental processes during cardiac development are regulated by proper cardiac contraction/blood flow-induced biomechanical forces[64,71–75]. Interestingly, we found that reducing cardiac contraction did not significantly affect the attachment of hematopoietic cells to the endocardium (Fig. 6F–J), suggesting that hematopoietic cell maintenance in the heart is at least partly independent of shear stress. This feature could provide one of the distinct characteristics between endocardial- and DA-derived HSPC production and might point to the role of the endocardium as a reserve hematopoietic tissue that is not influenced by mechanical forces.

Finally, whether endocardial-derived hematopoietic cells play distinct functions compared with hematopoietic cells from the DA or other vascular sources remains to be investigated. Our results suggest that the endocardium can give rise to a minor population of hematopoietic cells (~10% of *Tg(cd41:*GFP)⁺ cells in the CHT) that might have different cellular characteristics than other endothelial sources. Interestingly, we observed a higher number of endocardial-residing hematopoietic cells when primitive erythropoiesis is abrogated, suggesting that the endocardium might compensate for the loss of red blood cells by producing HSPCs/MEPs or promoting their proliferation to contribute to circulation. In addition, the endocardial-residing hematopoietic cells may play a local function

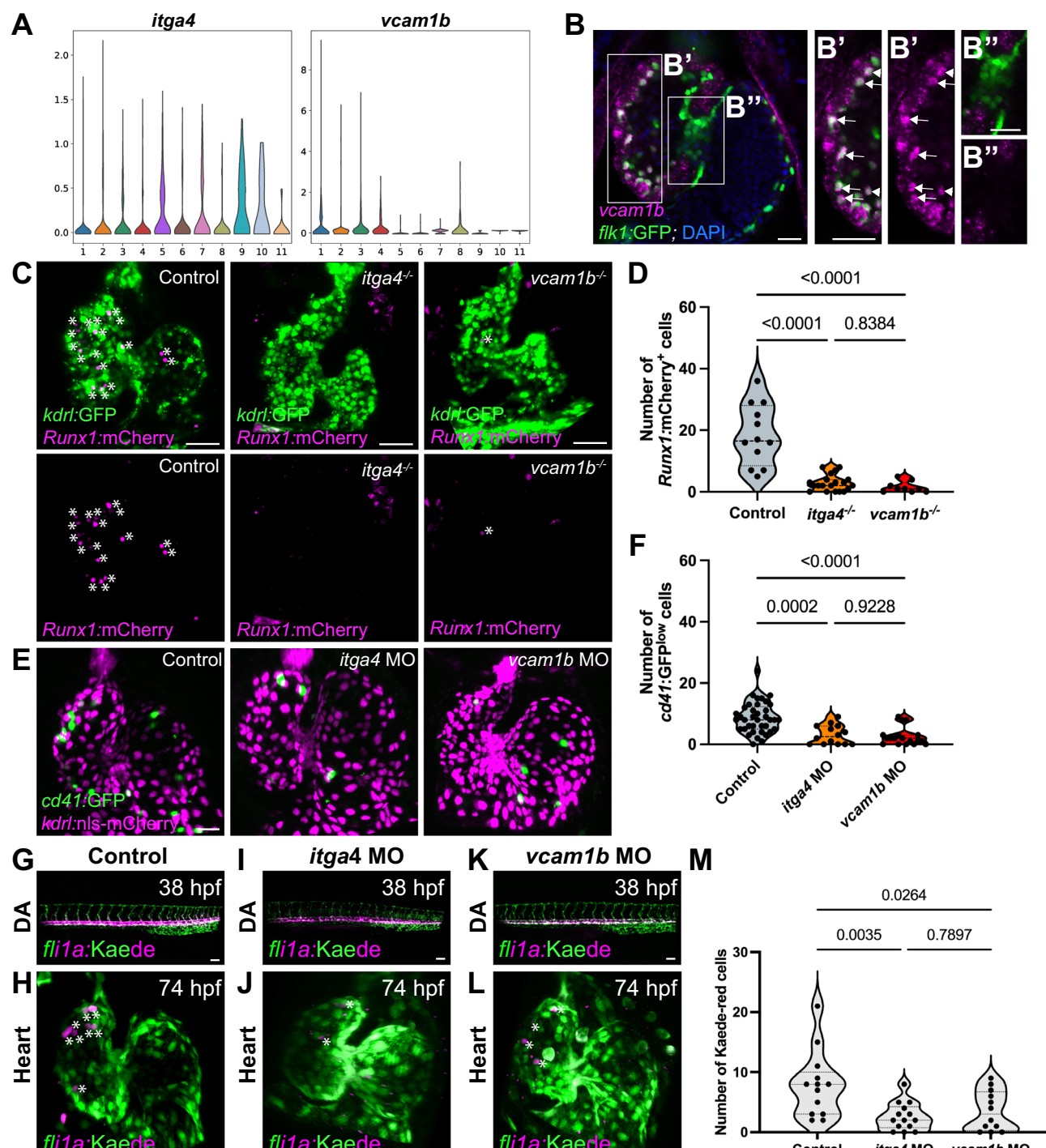

**Fig. 5 | HSPC attachment to the endocardium is dependent on Itga4-Vcam1b ligand-receptor interaction. A** Violin plots from the endocardial single-cell RNA-seq dataset show that *itga4* expression is enriched in the hematopoietic endocardial population (Cluster 5), whereas *vcam1b* expression is elevated in other endocardial cell populations. **B** RNAscope in situ hybridization for *vcam1b* mRNA in the 74 hpf zebrafish heart. *vcam1b* appears to be strongly expressed in the EdCs of the ventricular outer curvature (**B'**, labeled with *Tg(kdrl:*GFP) expression, white arrows), but appears undetectable in the valves (**B''**). **C** Representative confocal images of control, *itga4⁻/⁻*, and *vcam1b⁻/⁻ Tg(Runx1:mCherry), Tg(kdrl:GFP)* hearts at 74 hpf. White asterisks indicate *Runx1:*mCherry⁺ cells attached to the endocardium. **D** Significantly fewer *Runx1:*mCherry⁺ cells were found in *itga4⁻/⁻* and *vcam1b⁻/⁻* hearts compared with control. *n* = 12 control, 20 *itga4⁻/⁻*, and 9 *vcam1b⁻/⁻* hearts. **E** Representative confocal images of *Tg(cd41:GFP), Tg(kdrl:nls-mCherry)* hearts in

control, *itga4*, and *vcam1b* morphants at 74 hpf. **F** Quantification of *cd41:*GFP⁺ cells on the endocardium reveals a significant reduction in *itga4* or *vcam1b* morphants compared with control. *n* = 38 control, 14 *itga4*, and 18 *vcam1b* morphant hearts. **G, I, K** Representative confocal images of photoconverted endothelial cells in the middle and the posterior regions of the DA at 38 hpf in control (**G**), *itga4* morphants (**I**), and *vcam1b* morphants (**K**). **H** Photoconverted Kaede-red⁺ cells (white asterisks) are attached to the endocardium in 74 hpf control hearts. **J, L** Noticeably lower numbers of Kaede-red⁺ cells from the DA are present in 74 hpf hearts of *itga4* (**J**) or *vcam1b* (**L**) morphants. **M** DA-derived Kaede-red⁺ cell numbers in the heart are significantly reduced in *itga4* and *vcam1b* morphant larvae compared with control. *n* = 13 control, 14 *itga4*, and 12 *vcam1b* morphant hearts. **D, F, M** One-way ANOVA with Tukey's multiple comparison test was used. Scale bars, 30 μm (**B, C, E, G–L**). Source data are provided as a Source Data file.

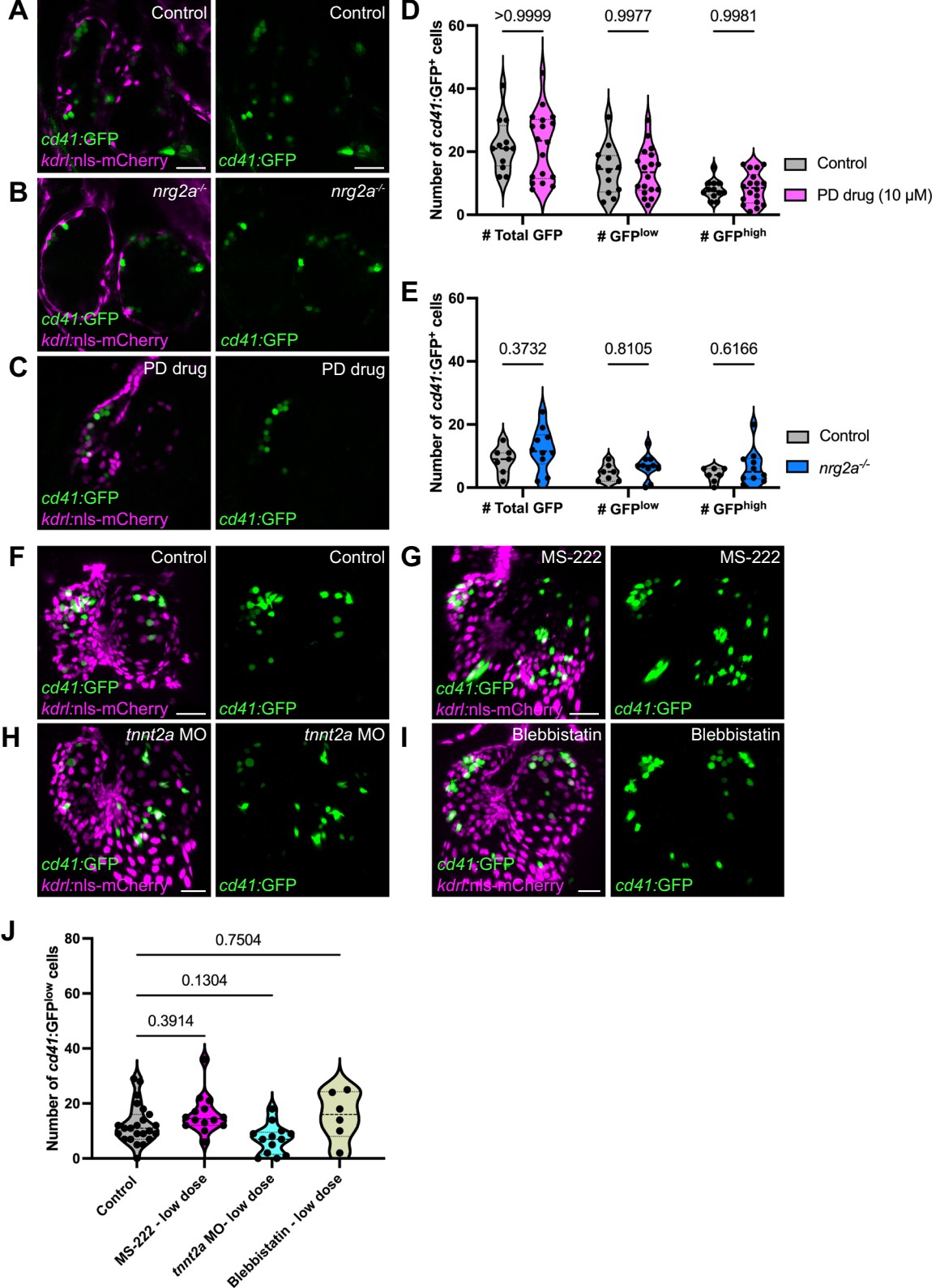

during homeostasis or injury. Consistent with this hypothesis, Shigeta et al.[39]. have reported that the endocardial-derived valve macrophages exhibit higher phagocytic activities, which is important to clear cellular debris in the tissue. In conclusion, we provide evidence that the endocardium contributes to hematopoiesis as both a tissue source and a niche, providing a unique model to study HSPCs under mechanical pressure.

## Methods

### Zebrafish handling and lines

Zebrafish were handled at the University of Münster, where the ethical guidelines outlined by the state of North Rhine-Westphalia were adhered to and the handling was conducted under the supervision of the veterinary authorities of the city of Münster, at the Max Planck Institute for Heart and Lung Research as approved by the Animal

**Fig. 6 | HSPC attachment to the endocardium is at least partly independent of cardiac trabeculation and contraction-induced biomechanical forces.**
**A–C** Representative confocal image of 74 hpf *Tg(cd41:GFP)*, *Tg(kdrl:nls-mCherry)* hearts. In *nrg2a*[−/−] mutants or larvae treated with the ErBB2 inhibitor PD168393 (PD), both leading to hypotrabeculation, *cd41*:GFP[+] cells remain attached to the endocardium. **D, E** Quantitative analysis reveals no significant difference in endocardial-attached *cd41*:GFP[+] cells in both models of hypotrabeculated hearts, PD drug treatment (**D**) or *nrg2a* mutants (**E**), compared with control. **D** n = 12 control, 18 PD-treated hearts; (**E**) 7 control, 10 *nrg2a*[−/−] hearts. **F–I** Suboptimal concentrations of MS-222 (**G**), *tnnt2a* morpholino (**H**), or Blebbistatin (**I**) were used to reduce cardiac contractions compared with control (**F**). No noticeable changes in *cd41*:GFP[+] cell numbers were observed. **J** Quantifications show no significant changes in *cd41*:GFP[low+]/HSPC/MEP numbers upon reduction of cardiac contraction. n = 23 control, 13 *tnnt2a* morphant, 6 Blebbistatin-treated, and 14 MS222-treated hearts. Two-way ANOVA with Sidak's multiple comparison test (**D**, **E**) or One-way ANOVA with Tukey's multiple comparison test (**J**) was used. Scale bars, 30 μm (**A–C**, **F–I**). Source data are provided as a Source Data file.

Protection Committee of the Regierungspräsidium Darmstadt, or at the University of Wisconsin-Madison in accordance with their Institutional Animal Care and Use Committee guidelines. All procedures performed on animals conform to the guidelines from Directive 2010/63/EU of the European Parliament on the protection of animals for scientific purposes.

Embryos were raised in a solution composed of 0.33 Danieau's medium, containing the following concentrations: 17.4 mM NaCl, 0.21 mM KCl, 0.12 mM MgSO₄·H2O, and 0.18 mM Ca(NO3)₂, while being maintained at a stable temperature of 28 °C. The following strains of zebrafish were maintained under standard conditions as previously described[76]: *Tg(kdrl:nls-mCherry)*[is4Tg,77], *Tg(kdrl:GFP)*[s843,56], *Tg(kdrl:BFP-CAAX)*[mu293Tg,78], *Tg(gata1:DsRed)*[sd2,79], *Tg(mpx:GFP)*[i114Tg,80], *Tg(itga2b/cd41:GFP)*[la2,79], *Tg(Runx1:mCherry)*[22], *Tg(fli1a:Gal4)*[ubs4]; *Tg(UAS:Kaede)*[rk8Tg,81], and *nrg2a*[mn0237Gt,66].

The *itga4* and *vcam1b* mutant lines were established at the University of Wisconsin-Madison via injection of CRISPR-Cas9 ribonucleo-protein complex in single-cell stage embryos. The following sgRNA for *itga4* and *vcam1b*, respectively, was used: 5′-TAAAAGCTGGAATTGGC CAC-3′ and 5′-GCGCGGCAGCTCCAGCGTGT-3′. This created a 13 base-pair deletion in exon 2 of the *itga4* gene resulting in a frameshift mutation, with the deleted sequence as follows: 5′-CCTGTGGCCAATT-3′, and a 7 base-pair deletion in the exon 4 of the *vcam1b* gene resulting in an early stop codon at amino acid 34, with the deleted sequences as follows: 5′-AATTTAA-3′. *itga4* and *vcam1b* mutant carriers were genotyped by PCR amplification and Sanger sequencing. The mutant embryos were raised in a solution composed of 1x E3 medium containing 5 mM NaCl, 0.17 mM KCl, 0.33 mM CaCl2, and 0.33 mM MgSO4 and supplemented with methylene blue to prevent fungal growth, while being maintained at a stable temperature of 28 °C.

### Confocal microscopy and image analysis
Confocal microscopes were employed for capturing images of arrested hearts. Embryos and larvae, up to 120 hpf, were embedded in 1% low-melting agarose supplemented with 0.2% Tricaine for immobilization. 7 and 10 dpf larvae were fixed in 4% paraformaldehyde for 2 h at room temperature prior to imaging, and then mounted in 1% low-melting agarose. Except for *itga4* and *vcam1b* mutants, images of stopped hearts and the CHT were acquired using a Zeiss LSM710 or Zeiss LSM880 confocal laser microscope with a 20x or 40x magnification dipping lens. For imaging stopped hearts in *itga4* and *vcam1b* mutant embryos, images were acquired using a Nikon A1 HD25 confocal laser microscope with a 25x magnification dipping lens set at 1.6x zoom. Maximal intensity projections were uniformly generated with identical settings applied to all samples. Image processing and subsequent quantitative analyses were conducted using Fiji (National Institutes of Health, USA) software tools and Imaris (Bitplane, UK). Brightness and contrast were adjusted with Imaris and Fiji. For imaging beating hearts, videos were acquired using a Zeiss Cell Observer spinning disk microscope at 100 frames per second with the 40X water immersion lens.

### Cardiac light sheet imaging and analysis
72 hpf *Tg(cd41:GFP)*[+], *Tg(kdrl:*nls-mCherry)[+] zebrafish embryos were mounted in 1.5% low-melting agarose (#A9414; Sigma) in FEP tubes (Pro Liquid). The inner and outer diameters of the FEP tubes were 0.8 mm and 1.2 mm, respectively. To immobilize the embryos while maintaining its physiological heartbeat, tricaine (MS-222) was supplemented to the agarose and the imaging chamber at a final concentration of 133 mg/l. This setup was used for time-lapse imaging in a custom-built light sheet microscope, featuring a Nikon 16×0.8 NA (CFI75 LWD 16X W) detection objective for high-resolution imaging. The system included a Hamamatsu Image Splitter (#A12801-01; W-View GEMINI) for simultaneous dual-channel fluorescence imaging and a pco.edge 4.2 CLHS sCMOS camera configured to capture images at 200 frames per second (fps) with a 5 ms exposure time per frame. Emission filters used were a BP525/25 bandpass filter for the green channel and an LP600 long-pass filter for the red channel. Excitation of fluorescent probes was conducted with a TOPTICA iChrome CLE laser system. Movie stacks of 10 z-planes in the central location of the heart were recorded for 1.5 s in each plane with 1 μm z-spacing. Time-lapse movies with a 30-minute interval were recorded for 7 h. Recorded movie stacks were synchronized as previously described[82]. Subsequently, cells expressing *Tg(cd41*:GFP) were manually visualized and tracked throughout the dataset.

### Localization analysis of *cd41*:GFP[+] cells in cardiac regions
For the precise determination of *cd41*:GFP[+] cell localization within cardiac tissues and their respective regions, Fiji software (National Institutes of Health, USA) was employed. A manual inspection of each z-stack derived from confocal images was undertaken. Subsequently, the heart was segmented into eight regions of interest, ensuring consistency across all analyzed samples. GFP-expressing cells with low intensity were enumerated within each predefined area. The resulting data were visually represented as a heatmap, displaying the percentage of cells within each region relative to the total *cd41*:GFP[+] cell count across all regions.

### Analysis of endocardial and *cd41*:GFP[+] surface area co-localization
A co-surface analysis was conducted to quantify the extent of co-localized tissue between the endocardium (*Tg(kdrl*:BFP-CAAX) expression) and *cd41*:GFP[+] cell populations. Imaris software (Bitplane, UK) was utilized for this purpose, facilitating the creation of surface renderings for the endocardial and *cd41*:GFP[+] cell populations. Subsequently, the shared surface area in μm³ and percentage of the total surface area was determined.

### Cell collection and flow cytometry
To isolate EdCs for the single-cell RNA sequencing experiment, hearts from 48- and 72-hpf *Tg(kdrl*:nls-mCherry)[+]; *Tg(cd41*:GFP)[+] larvae were manually dissected in DMEM + 10% FBS. Hearts were then centrifuged for 60 s at 3000 rpm, washed with 1 mL Hanks' Balanced Salt Solution and dissociated into single cells by incubating in 100 μl Enzyme 1 and 5 μl Enzyme 2 (Pierce Cardiomyocyte Dissociation Kit, Thermo Fisher Scientific, Cat#88281) for 20 min at 300 rpm in a 30 °C shaker[83]. The samples were centrifuged for 5 min at 3000 rpm, the supernatant was discarded, and fresh medium was added to the dissociated cells and passed through 40 μM-filter polystyrene 5 ml tubes. Negative controls of non-fluorescent hearts or single-color fluorescent hearts were

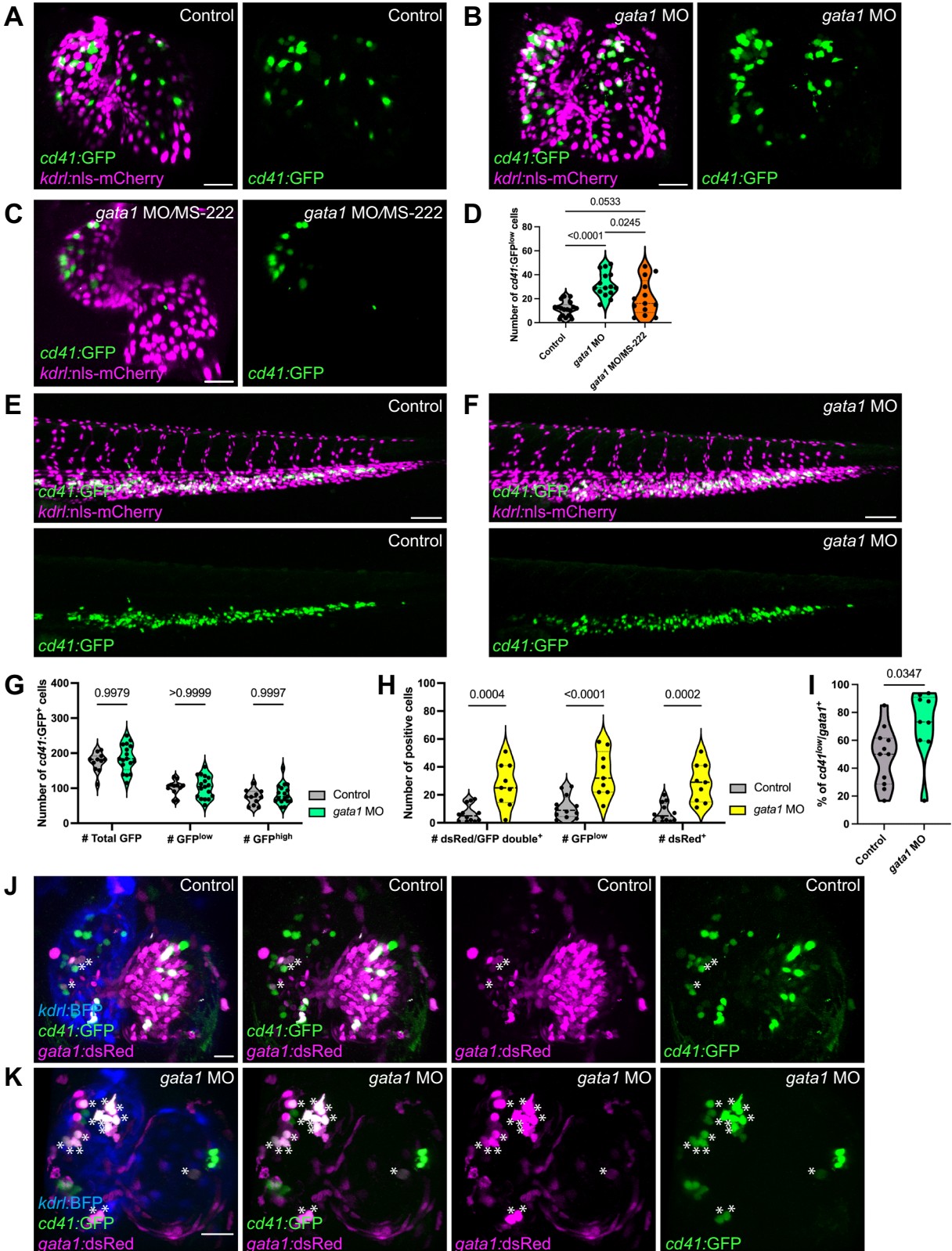

DMEM + 10% FBS. Cells from both collected samples were immediately combined and processed for the single-cell RNA sequencing. mCherry fluorescence was measured with 50 mW 561 nm excitation paired with 610/20 nm band pass filter. GFP fluorescence was measured with 50 mW 488 nm excitation paired with 530/30 band pass filter. Sorted cell data was recorded and analyzed on BD FACSDiva v8.0.1 software.

**Fig. 7 | Perturbing primitive erythropoiesis enhances the presence of cardiac-residing hematopoietic cells independently of blood flow.** **A**–**C** Representative confocal images of 74 hpf control (**A**) and *gata1* morphant (**B**) *Tg(cd41:GFP)*, *Tg(kdrl:nls-mCherry)* hearts, as well as *gata1* morphant *Tg(cd41:GFP)*, *Tg(kdrl:nls-mCherry)* hearts treated with MS-222 (**C**). Noticeable increase of *cd41*:GFP$^+$ cells present in *gata1* morphant hearts treated with DMSO (**B**) or MS-222 (**C**). **D** Quantification of *cd41*:GFP$^+$ cells in untreated or MS222-treated control and *gata1* morphants. Untreated *gata1* morphant hearts exhibited significantly increased *cd41*:GFP$^{low+}$ cell numbers compared with control, whereas MS-222 treated *gata1* morphant hearts exhibited a trend towards increased numbers of *cd41*:GFP$^{low+}$ cell numbers. *n* = 17 control, 15 *gata1* morphant, and 13 MS222-treated *gata1* morphant hearts. **E**, **F** Representative confocal images of control and *gata1* morphant *Tg(cd41:GFP)*, *Tg(kdrl:nls-mCherry)* CHTs at 74 hpf. **G** In contrast with the heart, total

numbers of *cd41*:GFP$^+$ cells did not change in the CHT upon loss of primitive erythrocytes in *gata1* morphants. *n* = 11 control and 17 *gata1* morphant CHTs. **H**, **I** Quantification of *cd41*:GFP$^{low+}$ and *gata1*:DsRed$^+$ cell numbers show a significant increase in the total numbers and proportions of double-positive cells upon *gata1* knockdown. (**H**) *n* = 12 control, 9 *gata1* morphant hearts; (**I**) *n* = 11 control, 9 *gata1* morphant hearts. **J**, **K** Representative confocal images of control and *gata1* morphant *Tg(cd41:GFP)*, *Tg(gata1:DsRed)*, *Tg(kdrl:BFP-CAAX)* hearts at 74 hpf. Asterisks indicate cells positive for both *Tg(cd41*:GFP), *Tg(gata1*:DsRed) expression. Noticeably higher numbers of double-positive cells were observed in *gata1* morphant hearts. One-way ANOVA with Tukey's multiple comparison test (**D**), two-way ANOVA with Sidak's multiple comparison test (**G**, **H**) or unpaired, two-tailed *t* test (**I**) was used. Scale bars = 30 μm (**A**–**C**, **J**, **K**), 100 μm (**E**, **F**). Source data are provided as a Source Data file.

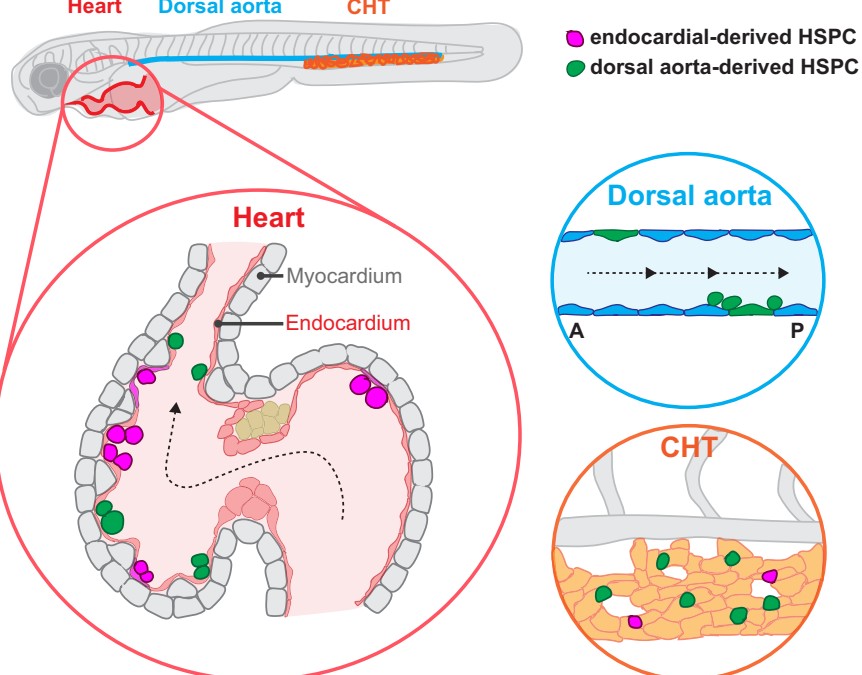

**Fig. 8 | Schematic model showing the heart as a resident tissue for and a contributor of hematopoietic stem and progenitor cells.** The endocardium in red and the myocardium in gray. Endocardial cells undergo EHT (flat magenta cells), leading to the formation of HSPCs (round magenta cells in the endocardium). Additional HSPCs originating from the DA, particularly in the posterior DA region,

attach to and stably integrate into the endocardium (green round cells). HSPCs originating from the endocardium and the DA contribute to the HSPC population in the CHT. Figure 8 created with BioRender.com released under a Creative Commons Attribution-NonCommercial-NoDerivs 4.0 International license.

## Single-cell RNA sequencing

For single-cell RNA sequencing, FACS-sorted cell suspensions were counted with Moxi cell counter and diluted according to the manufacturer's protocol to obtain a total of 2,660 cells at 48 hpf and 4,125 cells at 72 hpf. Each sample was run separately on a lane in Chromium controller with Chromium Next GEM Single Cell 3′ Reagent Kits v3,1 (10xGenomics). scRNA-seq library preparation was done using standard protocol, and sequencing was done on Nextseq2000. scRNA-seq analysis was performed as previously described[84] and final data visualization was done by CellxGene package (doi:10.5281/zenodo.3235020).

The RNA-sequencing data generated in this study have been deposited in the NCBI GEO database under accession code GSE269378.

## Photoconversion of endothelial cells

*Tg(fli:*Gal4)$^+$, *Tg(UAS:*Kaede)$^+$ larvae with the photoconvertible protein Kaede specifically expressed in endothelial cells were used for photoconversion. Specific vascular regions, which were the endocardium

or the DA, were selected for photoconversion using the Zeiss LSM710 confocal microscope. The photoconversion was accomplished through five scans of the 405-nanometer wavelength laser at 100% power. Z-stacks of Kaede-green and Kaede-red channels with the larval hearts before and after photoconversion were obtained to ensure spatially restricted photoconversion. Within the DA, we defined the anterior portion as the anterior-most part of the DA until just before the first intersegmental vessel (ISV) sprout (Fig. 4A), the "middle region" as the DA region adjacent to the first 12 ISV sprouts (Fig. 4D), and the "posterior region" as the posterior-most DA region from the 13th ISV sprout until the tail (Fig. 4G). Subsequently, the photoconverted embryos were retrieved and grown at 28 °C until subsequent imaging.

## Morpholino-mediated knockdown

Knockdown experiments were conducted by injecting specific antisense oligonucleotide morpholinos (MO) acquired from Gene Tools,

LLC. The MO solution, comprising a total volume of 1 nl, was injected into one-cell-stage embryos at the designated concentration:

*gata1* MO[68]: 5′-CTGCAAGTGTAGTATTGAAGATGTC-3′, final concentration used was 400 μM.

*itga4* MO[63]: 5′-CTCCAGTGAGTTGTGATGAAATCAT-3′, final concentration used was 100 μM.

*vcam1b* MO[63]: 5′-GTCAGCAAGAAAGCTGTATCCATGT-3′, final concentration used was 400 μM.

*tnnt2a* MO[85]: 5′-CATGTTTGCTCTGATCTGACACGC-3′, final concentration used was 100 μM for high and 10 μM for low concentration injections, respectively.

### EdU labeling

The Click-iT reaction for EdU staining was executed per the manufacturer's instructions (ThermoFisher #C10337). The samples were fixed in 4% paraformaldehyde, and incubated in Click-iT EdU Alexa Fluor 647 for 30 min. They were also incubated in anti-eGFP primary antibody (1:500 dilution, Thermo Fisher A-11122, polyclonal rabbit) overnight at 4 °C, and then Alexa Fluor-488 secondary antibody (1:500 dilution, Thermo Fisher A32723, goat secondary antibody) for 2 h at room temperature prior to imaging. Imaging was accomplished utilizing a Zeiss 710 confocal laser microscope, employing a 40x water immersion objective. Quantitative analysis of proliferative *Tg(cd41*:GFP)⁺ cells was conducted using the Imaris software (Bitplane, UK). Cells were classified as proliferative when they exhibited dual labeling with *Tg(cd41*:GFP) expression and EdU. The proportion of EdU-positive cells was calculated as the percentage of proliferative cells relative to the total count of *Tg(cd41*:GFP)⁺ cells.

### Fluorescent in-situ hybridization of *myb*

The following primers were used to create the in-situ probes for *myb*:

Forward: GCTGCTAAAGTCAGCCCAAC

Reverse:  <u>TAATACGACTCACTATAGGG</u>GTTTAATCGTGCCGACCACT

The PCR template was prepared using these primers and cDNA from 48 hpf embryos. The probe was then prepared by incubating the PCR template with the T7 RNA polymerase (Promega), 10X DIG NTP labeling mixture (Roche), transcription buffer (Promega), and RNase inhibitor (Promega) at 37 °C for 2 h and purified using the Zymo RNA Purification Kit. The purified RNA probe was then diluted in hybridization buffer in a concentration of 1 μg/μL. Fluorescent in situ hybridization was performed on 72 hpf *Tg(kdrl*:GFP)⁺ larvae. Larvae were fixed in 4% paraformaldehyde for 3 h at room temperature, then washed with RNase-free PBS/0.1%Tween for 1 h. They were permeabilized in serial methanol dilutions for 5 min each and incubated overnight in 100% methanol at -20 °C. They were rehydrated in PBS/Tween, incubated in 1 μg/mL Proteinase K for 30 min, and fixed again in 4% paraformaldehyde for 20 min. After washing with PBS/Tween, the larvae were incubated in hybridization buffer (Hyb buffer; 50% formamide, 5X SSC, 0.1% Tween-20, 50 μg/mL heparin, 500 μg/mL tRNA) at 65 °C for ≥2 h, and then in DIG-labeled RNA probe for *myb* (1 μg/mL in Hyb buffer) at 65 °C overnight. After serial dilutions with Hyb buffer/ SSC, the larvae were incubated in anti-tyramide-Alexa 568 (Thermo Fisher Scientific) in TSA buffer (100 mM Borate pH 8.5, 0.1% Tween-20, 2% Dextran Sulfate, 0.003% H₂O₂, 450 μg/mL 4-iodophenol) for 30 min at room temperature. The larvae were subsequently incubated overnight at 4 °C in anti-eGFP primary antibody (1:500 dilution, Thermo Fisher A-11122, polyclonal rabbit), and then Alexa Fluor-488 secondary antibody (1:500 dilution, Thermo Fisher A32723, goat secondary antibody) for 2 h at room temperature prior to imaging.

### RNAscope assay of *vcam1b*

Zebrafish embryos (*Tg(kdrl:GFP)*) at 73 h post-fertilization (hpf) were fixed overnight at 4 °C in 4% paraformaldehyde (PFA). Following fixation, embryos were rinsed with PBST (0.1% Tween-20 in PBS) and stored in 100% methanol at -20 °C. For rehydration, embryos underwent a series of ten 10-minute washes at room temperature (RT) with decreasing concentrations of methanol (75%, 50%, and 25%) in PBST, followed by a 10-minute wash in PBST. The RNAscope™ Multiplex Fluorescent Reagent Kit v2 (Advanced Cell Diagnostics, Cat. No. 323100) was used according to the manufacturer's instructions with several modifications. Embryos were incubated in 3% H₂O₂ for 10 min at RT, followed by washes with PBST, and then treated with RNAscope Protease III (Cat. No. 322337) for 5 min at RT and subsequently washed three times with 0.01% PBST (0.01% Tween-20 in PBS). A pre-hybridization step was included by incubating the embryos at 40 °C for 2 h in RNAscope probe diluent (Cat. No. 300041). The *vcam1b-C3* probe (Cat. No. 445681-C3) was then applied, and embryos were incubated overnight at 40 °C. All subsequent wash steps were performed with 0.01% Tween-20 in 0.2X SSCT, three times for 15 min each. Post-probe incubation, a postfix treatment with 4% PFA was conducted for 20 min at RT. All 40 °C incubation steps were performed in a water bath. Following the RNAscope protocol, embryos were washed three times with PBST for 5 min each. Blocking was performed for 1 h in a blocking solution (2% BSA in PBST). Embryos were then incubated overnight at 4 °C with a chicken polyclonal anti-GFP antibody (1:500; Invitrogen, Cat. No. A10262). After primary antibody incubation, embryos were washed six times for 30 min each with PBST. Secondary antibody incubation was carried out overnight at 4 °C using a goat anti-chicken Alexa Fluor 488 antibody (1:1000; Invitrogen, Cat. No. A11039). Post-secondary antibody incubation, embryos were washed six times for 30 min each with PBST. Nuclear staining was performed using DAPI. Before imaging, embryos were washed with 0.01% Tween-20 in PBS and mounted in 1% low melting point agarose (Invitrogen, Cat. No. 16520) in PBS for confocal imaging.

### Ex vivo cardiac culture

Cardiac extraction followed an established large-scale extraction protocol as described[86]. Leibovitz's L1-5 culture medium was enriched with 10% fetal bovine serum by a previously outlined protocol[61]. All procedural steps were conducted at room temperature, and the culture medium was pre-warmed to 28.5 °C. Subsequently, hearts were flushed from the filtration system into a petri dish coated with agarose and cultured at 28.5 °C for 24 h. Before confocal imaging, hearts were fixed with 4% PFA for 15 min and mounted in 1% low-melting agarose.

### Heartbeat quantification and reduction assay

To reduce heartbeat, we used low concentrations of *tnnt2a* MO, Tricaine (MS-222) or Blebbistatin treatment. For the *tnnt2a* knockdown, one-cell-stage embryos were injected with a low concentration of *tnnt2a* MO (0.1 ng/ul). Tricaine or Blebbistatin was added at a final concentration of 0.04% (w/v) or 1 μM, respectively, to the egg water from 48 to 74 hpf. To quantify their heart rates, live zebrafish embryos were observed under a stereomicroscope, and the heart rate was counted as the number of ventricular contractions per minute. Each condition was analyzed in triplicate with a minimum of 9 embryos per replicate.

### Treatment with PD drug (ErbB2 inhibitor)

*Tg(cd41:GFP), Tg(kdrl:nls-mCherry)* embryos were dechorionated and incubated in egg water supplemented with 10 μM PD168393 (dissolved in DMSO) from 50 to 72 hpf at 28 °C, and then immediately imaged. Control embryos were incubated in egg water containing an equal amount of the respective solvent (DMSO).

### Statistics and reproducibility

Unless otherwise specified, all experiments in this research study were conducted using a minimum of three independent biological replicates. Statistical analyses were conducted employing Graph Pad PRISM software. To compare two groups with normally distributed data, an

unpaired two-tailed Student's t-test was employed. For comparisons involving more than two groups, we utilized a one-way ANOVA with Tukey's multiple comparison test; in the presence of two categorical independent variables, a two-way ANOVA with Sidak's multiple comparison test was applied. A significance level of $\leq 0.05$ was employed for all experiments ($*p \leq 0.05$; $**p \leq 0.01$; $***p \leq 0.001$; $****p \leq 0.0001$).

Data from all experiments listed in Figs. 2–7, Figs. S2–8, S10–12 were obtained from at least three independent biological replicates. One exception is the *vcam1b* RNAscope assay in Fig. 5B, which was performed in two independent biological replicates using two different genotypes.

### Schematic model illustration
The model in Fig. 8 was drawn by free hand in Adobe Illustrator, from a template of Biorender images (licensed to D.B.).

### Reporting summary
Further information on research design is available in the Nature Portfolio Reporting Summary linked to this article.

## Data availability
The RNA-sequencing data generated in this study have been deposited in the NCBI GEO database under accession code GSE269378. Source data are provided with this paper.

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

## Acknowledgements

This work started in the laboratory of DYRS at the MPI-HLR in Bad Nauheim and, from 2022, continued in the laboratory of F.G. at the University of Münster. We thank Marlies Rossmann, Erez Raz, Arica Beisaw, Leah Biggs, Stefan Schulte-Merker, and Christian Mosimann for discussions and/or critical reading of the manuscript, and Sara Wickstrøm, Leah Biggs, and Radhan Ramadass for help with image analysis and quantification. Confocal imaging support was provided by the University of Wisconsin-Madison Optical Imaging Core. This work was supported by funds from the Add-on Fellowship of the Joachim Herz Foundation and the German National Academy of Sciences Leopoldina fellowship (LPDS 2021-01) to D.B., the Alexander von Humboldt Foundation (Humboldt professorship), and the DFG (Deutsches Forschungsgemeinschaft; Excellence Strategy Project EXC 2067/ 1-390729940) to J.H., the National Institutes of Health National Heart, Lung, and Blood Institute (R01HL174965, R01HL142998, R56HL142998), an American Society of Hematology Bridge Grant Award, and the Department of Cell and Regenerative Biology (University of Wisconsin-Madison) to O.J.T., the Max Planck Society and the European Research Council (AdG 101021349-TAaGC) to D.Y.R.S., and the DFG (Projects SFB1348B12 and SF1450N04), the Cells-in-Motion Interfaculty Center and the Faculty of Medicine (University of Münster; the Innovative Medical Research grant (I-GU122208)) to F.G.

## Author contributions

F.G. initiated the study in the group of D.Y.R.S. D.B. and F.G. developed the project and designed the research basis. D.B., A.V.H., L.S., N.M.W., and F.G. performed the majority of the experiments. P.M., A.G., A.A., O.S.L., and M.W. assisted with experiments and data analysis. K.K. provided technical assistance with the flow cytometry, and S.G. assisted with the single-cell RNA sequencing run and analysis. J.H., O.J.T., D.Y.R.S., and F.G. supervised and acquired funding for the project. F.G. and D.B. wrote the first draft of the manuscript with substantial input from D.Y.R.S. and O.J.T. All authors edited subsequent drafts of the manuscript.

## Funding

## Competing interests

The authors declare no competing interests.
