## [Peer Review File · Nature Communications]

The heart is a resident tissue for hematopoietic stem and progenitor cells in zebrafishEditorial Note: Parts of this Peer Review File have been redacted as indicated to remove third-party material where no permission to publish could be obtained.

REVIEWER COMMENTS

Reviewer #1 (Remarks to the Author):

The manuscript of Bornhorst et al, is describing the formation and the homing of hematopoietic stem cells (HSPCs) in the heart using zebrafish. Using confocal live imaging, cell tracing and scRNA seq analyses, the authors found that a heart's HSPCs can derive in part from the endocardial endothelia cells (EdCs) and part from the Aorta-gonad-mesonephros (AGM) transdifferentiation. The HSPCs homing in the heart required specialized adhesive EdCs but is non-dependent on heart trabeculation or heart mechanical forces. Finally, they provide evidence that a heart's HSPCs can produce erythrocytes possibly independently from primitive hematopoiesis or AGM HSPCs.

Overall, the experiments conducted are well quantified and carried within the standards in the field. Furthermore, the data presented has the potential to help clarify an important point in the hematopoiesis field related to the initial birth of HSPCs and new secondary hematopoietic organs in embryonic development. Hence the manuscript raises important and impactful points that are appropriate with the goal of the Nat Comm journal.

Here are the main comments that will be important to address for publication of the current data:

At the conceptual level the manuscript highlights two main discoveries: 1) that heart HSPCs can derive from EdCs, 2) that heart HSPCs can also derive from AGM, thus the heart can function as a secondary hematopoietic organ. Both points are important and valid. In essence, the paper will not suffer for having one scenario or the other. However, these two points should be better documented in the current manuscript.

Figure 2 uses cell tracing experiments. This is the figure where points 1 and 2 can be clarified better:

1) They should provide the quantification of Kaede cell at 74, 96 and 120 hpf in the heart besides the one already quantified in the CHT. From the images it looks like most of the Kaede+ cells come from the AGM conversion (see white stars in 2C and D). The authors say 4 cells on average but the same cells if not less seems to be present in the heart after photoconversion on the EdCs. Can these cells be quantified at 74 but also later stages?

2) Photoconversion of the AGM at 22 hpf is quite early for the actual EHT (24-32hpf pick until 48 hpf). It could also be early for the EdCs considering that the first heart HPSC appears 48 hr later. Can the author provide photoconversion analysis of AGM at different time points? e.g. AGM, EdC at 26, 32, 48 hpf. This will help to consolidate the proportion of AGM vs EdC derived HSPCs which is an important point for this paper.

3) Can the authors quantify the Kaede positive cells not only in the CHT but also in the Thymus (T) at 74 hpf? The prediction would be that the EdC derived HSPCs will not lead to Kaede + cells in the T but only CHT over 74 and ~96 hpf due to the bias in Erythroid/myeloid lineages.

4) One important conceptual clarification that the authors should provide is whether AGM derived HSPCs reside in the heart after delamination and become erythroid/myeloid biased in this heart. Alternatively, another scenario is that HSPCs are born erythroid/myeloid biased in the AGM and then nice in the heart for proliferation. Another scenario is that HSPCs derived AGM are not lineage biased, but the EdC derived HSPCs are. Therefore, some HSPCs in the heart will convert in erythroid/myeloid other in all lineages. The author should provide experiments to dissect which

scenario is more likely to happen. As mentioned in the introduction there are several controversies in this research topic, hence clarification on the homing vs transdifferentiation effects of EdCs will help to strengthen the significance of the findings.

5) It would be convincing to use the Kaede approach to repeat the experiments in figure 5 and consolidate the evidence that HSPC derived EdCs are producing Gata 1+ cells. One way to label the red blood cells without using the transgenic line gata1 is to use the antibody TER-119 (from zfin). This way the authors can see whether Kaede cells are TER-119 in the CHT with or without Gata-1 mo.

At the technical level the analysis of scRNA-seq could be further deepened to extract form information for the heart HSPCs.

1) I would suggest combining the cells from 2 and 3 dpf and reanalyzing the clusters formation. Having two time points in the same UMAP will enable you to perform cell projection analysis that might be able to observe EHT. Ideally the author could combine an additional scRNA seq experiment from an earlier time point if possible. Nevertheless, the results should be achievable with two time points only. RNA velocity, pseudotime or CellRank can be used to then perform the analysis of cell trajectories.

2) For the analysis above, the authors should verify i. is there transition between gata2, runx1, cmyb? ii. is there a transition from runx1 + and cmyc/gata1+ or cmyb/lcp1 (or spi1b)?

3) Could it be possible to perform the analyses from points 1 and 2 (above) by sub-clustering cluster 5? Hopefully the authors could have a better resolution of HSPCs with erythroid and myeloid bias and learn if they originate from EHT transition or not.

Reviewer #2 (Remarks to the Author):

This study shows that the endocardial cells in the zebrafish differentiate into hematopoietic cells. Authors used both in vivo high-resolution imaging and photoconversion tracing methods, as well as ex vivo heart culture experiments, to demonstrate the de novo contribution of endocardial cells to hematopoietic cell lineages. Mechanistically, integrin alpha 4 and Vcam1 regulates the attachment of these hematopoietic cells to endocardium. This work is important to the field of hematopoiesis, as endocardial contribution to HSC is a contentious topic with heated debate in the past few years. Elucidation of this issue is critical to our understanding of hematopoiesis in the development and also potential role in cardiac development/valve formation. The study is well-down and major conclusion is supported by the most data. I have a few concerns and questions for authors to clarify.

This study showed that the hematopoietic-promoting genes are enriched in non-valve endocardial cells. However, Shigeta et al., used genetic lineage tracing to demonstrate that hemogenic endocardium is a de novo source of tissue macrophages in the endocardial cushion, which is the primordial part for cardiac valves. Could this be due to difference of species or technical method used for two studies? Any explanation for this difference in heterogenous endocardial population in generating hematopoietic cells?

It is shown that the majority of HSPC were generated in the outer surface part of the ventricle close to the outflow canal. Any possible explanation for why such regional enrichment of endocardium-derived HSPC?

Recent study using conventional Cre-loxP genetic lineage tracing showed that endocardial cells minimally contribute to cardiac macrophages and hematopoietic cells in the circulation (Liu et al., JCB 2022). Additionally, intercellular genetic tracing demonstrates no contribution of endocardial cells to hematopoietic cells in the developing heart (Liu et al., Dev Cell 2023). These studies argue against the presence of hematopoietic endothelial cell population in the developing endocardium. Here the evidence of endocardial cells to generate HSPC in zebrafish is contradictory to these recent findings in mouse. Is this due to difference between species or approaches for studying cell lineages among different studies?

In *Itga4* or *Vcam1b* MO zebrafish, could EHT in the dorsal aorta also be impaired in addition to reduced attachment of hematopoietic cells to endocardium?

Is there any defect in valve formation in the mutant?

Reviewer #3 (Remarks to the Author):

Comments for Authors

The study performed by Bornhorst et al was aimed, using the zebrafish embryo/larva model, at addressing the potential differentiation of the heart endothelium (the endocardial tissue) into hematopoietic stem and progenitor cells (HSPCs) and of the endocardium to serve as a specific niche to ensure the maintenance of HSPCs. To identify molecular signatures of putative HSPCs lodging in the developing heart, they performed single-cell RNAseq using previously established double transgenic zebrafish lines expressing fluorescent reporters into vascular and hematopoietic cells and identified a population of HSPC candidate cells. This population appears to express markers of well characterized aorta-derived HSPCs generated via endothelial-to-hematopoietic transition (including *gata2b*, *runx1* and *myb*) and to more differentiated progenies (including *mpx* and *mpeg1.1* that are more specific of differentiated myeloid progenies, as well as *gata1* and erythrocyte functional markers). Using photoconversion of the protein Kaede expressed in the entire vascular system, they address the origin of heart HSPCs, after photoconversion of either the entire endocardium, or the dorsal aorta (the site of HSC production). To address the autonomous potency of the heart to produce HSPCs, the authors also cultivated the entire organ in vitro for approximately two days. They then perform functional studies aimed at identifying essential adhesion molecules involved in HSPCs maintenance and interactions with heart niche cells, namely and respectively the integrin *itga4* and the cell-adhesion molecule *vcam1b*. Finally, they provide evidence for the possible compensation by heart HSPCs of the megakaryocyte-erythroid specific lineage (MEPs) of red-blood cell (RBCs) loss after the impairment of RBC production by primitive erythropoiesis, by knocking down the transcription factor *gata1*.

The contribution of the developing heart in the de novo formation of HSPCs remains a debated question in vertebrate models (as mentioned in the authors introduction, although recent evidence in the mouse speaks against immune and blood cells being derived from the heart endothelium, see the matter arising by Liu et al Dev Cell 2023, Ref 34 in the authors ref list).

To tackle this important issue, Bornhorst et al use the power of the zebrafish embryo model to trace, in the entire animal, cell populations after photoconversion (here, for example, from the heart to the first hematopoietic niche (the caudal hematopoietic tissue – CHT) and from the dorsal aorta (the site of the endothelial-to-hematopoietic transition) to the heart, the latter for questioning the fact that heart HSPCs would only derive from the dorsal aorta). While photoconversion allows, in theory, to precisely address the organ/tissue/single-cell from which a given cell population originates, it requires a very precise methodology for accuracy. In addition, precise delimitation of specific tissue territories to photoconvert needs to be applied to prevent biases in interpretation. These two points are critical and, based on that, I believe that the work performed by Bornhorst et al did not reach the status of providing solid results in favor of de novo formation of HSPCs from the heart endothelium (see the details below). However, the data illustrating the idea of the heart endothelium being a specific niche for HSPC sub-types, rather leading to megakaryocyte-erythroid precursors (MEP), are more convincing (although requiring support with supplementary information on knocking down experiments using morpholinos).

Comments on Figures and Text:

1- Figure 1 and S1.

- The genes that characterize cluster 5; more genes should be given (and perhaps an Excel file as Supplemental with top 10-20 top genes at least for all clusters); looking at Suppl. Fig1B, the heat map is not informative, in particular for cluster 5 for which only cytochrome c oxidase (cox) subunits are listed, which are mitochondrial genes (and in general in sc-RNAseq is an indicator of dying cells); can the authors comment on that ?

In addition, the authors need to be careful in the text of the first paragraph of Results and not speak of 'genes that promote hematopoiesis' when they refer to enriched genes in cluster 5; *mpx* and *mpeg1.1* are functional markers of neutrophils and macrophages and are expressed in differentiated myeloid lineages. Hence cluster 5 appears to be very heterogeneous, with more or less differentiated cells of myeloid and erythroid lineages (could this cluster be sub-clustered ?). Each cluster as pointed in Suppl. Fig1C should be associated with a given heart-endothelial or hematopoietic cell type population to help the reader for figure analysis. This should be added to Figure 1 and not only to Suppl. Fig1.

Also, Suppl. Fig1D, owing to the importance of *runx1* in hematopoietic populations, the panel should show in which clusters of the UMAP it is enriched, is it in cluster 5 ? and what for *myb* and *drl* ?

- The quantification of co-localization Figure 1H, I (+ Methods); the co-localization of endothelial surface with CD41+ cells (H) and % co-localization (I), this cannot be accurate because endothelial cells are labelled with nls-mCherry that is targeted to the nucleus; hence, contacting surface cannot be determined (because this requires segmentation, hence cell contours). This comment is also valid for Suppl. Figure 4 (B and C).

To support Figure 1 (in particular panel G but also for a reader less acquainted with heart morphology and 3D shape and connecting veinous/aortic components), a representation of the heart and afferent/efferent vascular systems should be provided (in Supplemental for ex.). This is important in regard to the outflow tract, for reasoning in regard to increase in % positional mapping of CD41+ cells; I should guess it is contacting tissue of aortic nature ? if yes, it is of relevance in regard to photoconversion experiments as well as in regard to heart dissection for tissue culture (and potential 'contamination' with endothelium of aortic nature, see below comments on Figure 2).

2- Figure 2.

- The photoconversion is not precise. This is exemplified by the panel B that shows photoconversion of the dorsal aorta (DA) at 22hpf; one can clearly see that the bottom parts of inter-segmental vessels have been photoconverted, the aortic wall on the left side of the image only, and a portion of the sub-aortic space. In this situation, the quantitative outcome on HSPCs that will have emerged from the hemogenic endothelium (in the aortic floor) will be minimized in comparison of photoconversion of the whole endocardium (and hence the quantifications in panel F and G make no sense). More precise photoconversion of the hemogenic endothelium (the aortic floor) should be performed. In addition, the reader is missing the information on which region of the DA has been photoconverted (anterior part of the trunk ? entire DA in the trunk ? DA above the CHT region ?). If, as studied in the last part of this paper, part of the heart population of HSPCs that have important physiological functions come from the one or the other of these different regions, more rigorous photoconversion of these different DA regions should be performed.

Regarding the photoconversion of the endocardium, the reader should see a Z-projection to be able to appreciate precisely the areas that, indeed, have been photoconverted (also, the scales in panels A and B are very different). In addition, the cells that are spotted by the white asterisks are not convincingly positive for photoconverted Kaede (panel C, 74hpf, right). Finally, the CHT is full of pigment cells that can be easily mistaken for fluorescent cells (in the red and green channels); the authors should be cautious (see panel E) and may provide images of non-photoconverted animals. An alternative is to perform spectral deconvolution to eliminate pigment cell background.

- More precision should be added to the study by performing photoconversion of different areas of the heart endothelium. In link with the positional mapping of the CD41:egfp+ cells Fig 1G and the bias in the region nearby the outflow tract, could that region be more productive for potential de novo HSPC formation ? (if that makes sense for photoconversion at 22 hpf)

- Heart dissection and in vitro culture is a nice experiment to address the autonomous production of HSPCs by the endocardium. However, addressing this point by using the CD41:eGFP fish line may be problematic because the level of eGFP is very variable in this line (not to speak of mosaicism which is always at risk). Hence, there may be HSPCs that are not detectable at 48hpf and become visible at 74hpf.

In addition, it is difficult to appreciate if part of connecting arterial tissue may have been dissected together with the 48hpf heart; is that possible ? also, if the heart is dissected, it may respond to this physiological insult by awaking arterial functions, could this happen ? Probably, a more detailed analysis, using in situ hybridization for example, of the hemogenic status of the heart/heart endothelium should be performed (this addresses endogenous gene expression which prevents

from fluorescent Tg line-induced biases. Example of genes that could be tested: *gata2b*, *runx1*).

3- Figure 3

The characterization of *itga4* and *vcam1b* morpholinos: the authors should provide data showing the efficiency of the MO at the mRNA or protein levels. Images of embryo/larva should also be provided to show that they can be compared with wild type animals.

The results with the *itga4* mutant line: the effect looks very drastic on the heart morphology, even for heterozygous (panel E, left, eGFP signals for heart endothelial cells). In addition, the *runx1:mCherry* labelling is very odd (above the white asterisks), what are we looking at ?

4- The argument on mechanical force and its incidence on endocardial cell formation, as well as attachment (page 8). The text is somehow confusing because jumping from referring to attachment of HSPCs to the endocardium (not impaired by hypo-trabeculation, hence stronger shear stress, nor by biomechanical forces from flow) to endocardial hematopoietic cell formation (that hypothetically also does not require mechanical force). This is a major issue of the paper: the part that the authors attribute to HSPC formation from the endocardium versus attachment (niche aspects).

What would be the vision of the authors on the contribution of *runx1* in this context ? (whose expression is dependent on blood flow, in the context of the endothelial-to-hematopoietic transition in the aortic floor)

In conclusion, this paper reports more on the attachment/residence of HSPCs and the relationship to the heart cell niche than the formation de novo of HSPCs from the endocardium.

12 June 2024

We thank the reviewers for their constructive feedback, which we believe has helped to greatly improve our manuscript. Whenever technically possible, we have addressed all points raised by the reviewers.

In the new manuscript, we have now:

- (1) performed new and extensive photoconversion experiments and quantitative analysis of endothelial cells in the heart and dorsal aorta – particularly in identifying the regions and developmental timepoints in which the endocardium and the dorsal aorta contribute to hematopoietic cells in the heart, the caudal hematopoietic tissue, and the thymus;
- (2) performed imaging of the beating heart in 3D over a 7-hour period using cutting-edge, custom-built light sheet microscope (collaborative work at the University of Göttingen) to identify endocardial cells activating the HSPC marker;
- (3) conducted sub-clustering analysis of the endocardial-hematopoietic cluster to identify subpopulations of cardiac hematopoietic cells;
- (4) included new results of RNAscope *in situ* hybridization to show that *vcam1b*, which is known to be expressed in endothelial cells of HSPC niches, is expressed in endocardial cells in the ventricular outer curvature, where HSPCs preferentially adhere to;
- (5) obtained new images and quantitative analysis of cardiac-residing HSPCs in *vcam1b* mutants, which were not included in the previous version of the manuscript;
- (6) extended the photoconversion analysis in *itga4* and *vcam1b* morphants to provide evidence that their loss led to failure of the endocardium to retain circulating hematopoietic cells; and
- (7) have included a schematic model of our conclusions as the last figure.

We have added the following new Figures, Movies and Tables: new Figures 1B, 1I, 1J, 2A-J, 3A-J, 4A-M, 5B-D, 5G-M, 8, S1D-I, S2A-I, S4A-C, S6A-B, S7A-H, S8A-E, S9A-B; Videos 3-7.

All major changes in the manuscript are highlighted in yellow.

We believe that the results from the bulk RNA-sequencing of valve vs. non-valve endocardial cells do not add to the primary focus of the manuscript, and as such, have removed it (old Figure S3).

Please find below our point-by-point response to the reviewers.

Sincerely,

Felix Gunawan

REVIEWER COMMENTS

Reviewer #1 (Remarks to the Author):

The manuscript of Bornhorst et al, is describing the formation and the homing of hematopoietic stem cells (HSPCs) in the heart using zebrafish. Using confocal live imaging, cell tracing and scRNA seq analyses, the authors found that a heart's HSPCs can derive in part from the endocardial endothelia cells (EdCs) and part from the Aorta-gonad-mesonephros (AGM) transdifferentiation. The HSPCs homing in the heart required specialized adhesive EdCs but is non-dependent on heart trabeculation or heart mechanical forces. Finally, they provide evidence that a heart's HSPCs can produce erythrocytes possibly independently from primitive hematopoiesis or AGM HSPCs.

Overall, the experiments conducted are well quantified and carried within the standards in the field. Furthermore, the data presented has the potential to help clarify an important point in the hematopoiesis field related to the initial birth of HSPCs and new secondary hematopoietic organs in embryonic development. Hence the manuscript raises important and impactful points that are appropriate with the goal of the Nat Comm journal.

We thank the reviewer for their supportive comments.

Here are the main comments that will be important to address for publication of the current data:

At the conceptual level the manuscript highlights two main discoveries: 1) that heart HSPCs can derive from EdCs, 2) that heart HSPCs can also derive from AGM, thus the heart can function as a secondary hematopoietic organ. Both points are important and valid. In essence, the paper will not suffer for having one scenario or the other. However, these two points should be better documented in the current manuscript.

We thank the reviewer for this critical comment, and have performed extensive follow-up to the photoconversion and *ex vivo* transplant experiments, as well as adding new results from light sheet imaging of the beating heart, to strengthen our findings. These new results are included in new Figures 2A-J, 3A-J, 4A-M, S6A-B, S7A-H, and S8A-E.

Figure 2 uses cell tracing experiments. This is the figure where points 1 and 2 can be clarified better:

1) They should provide the quantification of Kaede cell at 74, 96 and 120 hpf in the heart besides the one already quantified in the CHT. From the images it looks like most of the Kaede⁺ cells come from the AGM conversion (see white stars in 2C and D). The authors say 4 cells on average but the same cells if not less seems to be present in the heart after photoconversion on the EdCs. Can these cells be quantified at 74 but also later stages?

In connection with this point by Reviewer 1 and as suggested by Reviewer 3, we extended our previous analysis by performing Kaede photoconversion experiments not only in the entire dorsal aorta, but also in spatially confined regions within the DA endothelium (anterior-most, middle yolk extension, and posterior-most regions). These new experiments uncovered the posterior-most and middle yolk extension as the DA regions that gave rise to the highest numbers of endocardial-residing hematopoietic cells (6 and 2.25 cells, respectively). Quantification of endocardial-residing cells at 74 hpf derived from the entire photoconverted DA also uncovered a higher number than the one we previously reported – an average of 9.75 cells, which account for roughly half (42%) of the numbers of *Tg(cd41:GFP⁺)* cells in the heart. Strikingly, when we imaged and quantified the number of cardiac-residing cells at 98 hpf, we found the average number (21 cells) had approximately doubled from the one at 74 hpf, accounting for approximately 63% of *Tg(cd41:GFP⁺)* cells in the heart. The increase in this number between 74 and 98 hpf could be due to a high degree of proliferation of DA-derived EdCs or attachment of more circulating cells to the endocardium. Thus, our new results show that the DA contributes a more significant proportion of the hematopoietic cells in the heart than we had estimated in our previous version of the manuscript. These results are now included in new Figure 4A-M.

Regarding quantification of endocardial-derived Kaede-red cells in the endocardium, we found it impossible to quantify because the entire endocardium has been photoconverted to the Kaede-red signal. However, we performed region-specific photoconversion of the endocardium (atrium, ventricular outer curvature, and ventricular inner curvature) at 48 hpf and imaged the same hearts 24 and 48 hours post conversion. Interestingly, we found Kaede-red cells in endocardial regions that were different than the original site of photoconversion. We found Kaede-red cells derived

from the atrium that were eventually attached to the ventricular outer curvature 48 hours post conversion in 9 out of 14 hearts examined (average of 1.9 cells per heart). These results indicate that atrial endocardial cells can extrude and attach to the ventricle, contributing to the hematopoietic cell population residing in the ventricular outer curvature. They are now included in new Figures 2G-J, S7A-H, and S8A-E.

2) Photoconversion of the AGM at 22 hpf is quite early for the actual EHT (24-32hpf pick until 48 hpf). It could also be early for the EdCs considering that the first heart HPSC appears 48 hr later. Can the author provide photoconversion analysis of AGM at different time points? e.g. AGM, EdC at 26, 32, 48 hpf. This will help to consolidate the proportion of AGM vs EdC derived HSPCs which is an important point for this paper.

We thank the reviewer for their comment. We chose to perform the photoconversion experiment at 22 hpf, a timepoint slightly earlier than the initial reported EHT event in the DA, because we reasoned that the endothelial cells budding off at the later timepoints (24-48 hpf) would still have the photoconverted Kaede-red proteins that are known to be stable for several days. Thus, we would minimize the possibility of missing any cells that undergo EHT in the DA. Nevertheless, as suggested, we repeated the photoconversion of DA endothelial cells at 38 hpf and imaged the heart at 74 hpf. We confirm that the DA still contributes to endocardial-residing cells when photoconverted at 38 hpf (7.85 cells), at numbers that are mostly comparable with DA photoconverted at 22 hpf (9.75 cells). Our results also suggested that the DA contributes to endocardial-residing cells mostly starting from 38 hpf onwards. These results are now included in new Figure 5G, M.

3) Can the authors quantify the Kaede positive cells not only in the CHT but also in the Thymus (T) at 74 hpf? The prediction would be that the EdC derived HSPCs will not lead to Kaede + cells in the T but only CHT over 74 and ~96 hpf due to the bias in Erythroid/myeloid lineages.

The reviewer raised an interesting suggestion. We photoconverted EdCs at 22 hpf, and then imaged the thymus at 74 hpf. We found endocardial-derived photoconverted red cells in the thymus, but at much lower numbers compared to the CHT. Quantification revealed an average of about 0.93 cells in the thymus, compared with 16.7 cells in the

CHT. In addition, we found endocardial-derived Kaede-red cells in all CHT we examined, but only in 57% (8 out of 14) thymus we imaged. These results suggest that the bias in erythroid/myeloid lineage of endocardial-derived hematopoietic cells leads to a preferential retainment of these cells in the CHT instead of the thymus, and are now included in new Figure 2D, F.

4) One important conceptual clarification that the authors should provide is whether AGM derived HSPCs reside in the heart after delamination and become erythroid/myeloid biased in this heart. Alternatively, another scenario is that HSPCs are born erythroid/myeloid biased in the AGM and then niche in the heart for proliferation. Another scenario is that HSPCs derived AGM are not lineage biased, but the EdC derived HSPCs are. Therefore, some HSPCs in the heart will convert in erythroid/myeloid other in all lineages. The author should provide experiments to dissect which scenario is more likely to happen. As mentioned in the introduction there are several controversies in this research topic, hence clarification on the homing vs transdifferentiation effects of EdCs will help to strength the significance of the findings.

Although Reviewer 1 has raised a very interesting question, we find it technically impossible to address this question with the available imaging technologies due to difficulties in following the dorsal aorta and the beating heart in the same animal and with the available lineage tracing tools due to lack of DA- or endocardial-specific Cre lines. Although Kaede photoconversion is very valuable in pinpointing the sources of hematopoietic cells residing in the heart, significant limitations due to fixation artifacts and photobleaching of fixed samples prevent us from performing immunostaining in Kaede-positive animals (see our response below, Figure R1). We have added this technical limitation in the Discussion section (second paragraph).

Figure R1. Fixation of Kaede protein leads to artifacts, and Kaede-red/Kaede-green can not be distinguished. (A) Representative live embryo with the dorsal aorta photoconverted at 24 hpf showing a clear Kaede-red photoconverted region. (B) Fixed embryos which have been photoconverted in the dorsal aorta region at 24 hpf visualize the Kaede-red signal in the whole tissue and the photoconverted region can not be identified. (C) Control embryos that have not been photoconverted show Kaede-red signal after fixation in the whole sample indicating a fixation artefact leading to Kaede-red signal.

5) It would be convincing to use the Kaede approach to repeat the experiments in figure 5 and consolidate the evidence that HSPC derived EdCs are producing Gata 1+ cells. One way to label the red blood cells without using the transgenic line gata1 is to use the antibody TER-119 (from zfin). This way the authors can see whether Kaede cells are TER-119 in the CHT with or without Gata-1 mo.

We thank the reviewer for this comment. We attempted to perform the immunostaining of animals that have been photoconverted, as suggested. However, we found that upon fixation, the entire vasculature appeared to become fluorescent in red, which we presume to be a strong increase in background fluorescence. We were unable to distinguish between the cells expressing the real photoconverted Kaede-red protein and the cells that are auto-fluorescent; hence, we cannot address the question whether DA-derived Kaede red cells are positive for TER-119 or other erythrocyte markers through immunostaining (Figure R1).

At the technical level the analysis of scRNA-seq could be further deepened to extract form information for the heart HSPCs.

1) I would suggest combining the cells from 2 and 3 dpf and reanalyzing the clusters formation. Having two time points in the same UMAP will enable you to perform cell projection analysis that might be able to observe EHT. Ideally the author could combine an additional scRNA seq experiment from an earlier time point if possible. Nevertheless, the results should be achievable with two time points only. RNA velocity, pseudotime or

CellRank can be used to then perform the analysis of cell trajectories.

2) For the analysis above, the authors should verify i. is there transition between *gata2*, *runx1*, *cmyb*? ii. is there a transition from *runx1* + and *cmyc/gata1*+ or *cmyb/lcp1* (or *spi1b*)?

3) Could it be possible to perform the analyses from points 1 and 2 (above) by sub-clustering cluster 5? Hopefully the authors could have a better resolution of HSPCs with erythroid and myeloid bias and learn if they originate from EHT transition or not.

We thank the reviewer for these 3 suggestions. We have now performed the subclustering of the endocardial-hematopoietic cluster (EHC; Cluster 5) to probe for specific hematopoietic populations within the cluster. Out of the 390 cells, we could distinguish a macrophage subcluster (24 cells) and a subset of cells (15 cells) that express high levels of neutrophil markers (*mpx* and *lyz*). The majority of cells in the EHC cluster expresses high levels of HSPC markers, including *myb*, *runx1*, and *gata2*. However, we could not clearly distinguish states of HSPC differentiation based on the gene expression of the subclusters. We also performed RNA velocity (Figures R2, R3), pseudotime and CellRank but could not obtain a clear biological significance from these analyses. We did not detect any endocardial cells that might be in transition into hematopoietic cells or are actively undergoing EHT. We speculate that this effect is likely due to the low numbers of EdCs that undergo EHT, and that 72 hpf might be too early to detect a sufficient number of EdCs that undergo EHT. Nevertheless, our experiments with the *ex vivo* transplant of hearts (new Figure 3A-C) and longitudinal light sheet recording of the beating heart in 3D (new Figure 3E-J) indicate that some endocardial cells can undergo EHT and activate the expression of HSPC markers.

Figure R2. RNA velocity and pseudotime analysis of the endocardial clusters. RNA velocity (A) and velocity-stream (B) analyses of the endocardial clusters do not reveal any clear connection between the endocardial-hematopoietic cluster (Cluster 5 in purple) and the other endocardial clusters.

Figure R3. RNA velocity and pseudotime analysis of the hematopoietic endocardial cluster (Cluster 5). (A) RNA velocity analysis of the subclusters in Cluster 5 does not uncover significant biological insides. (B) RNA velocity-stream analysis shows a flow from the center population to both edges; however, a clear biological function cannot be uncovered. (C) RNA velocity-pseudotime analysis does not give additional insides cell trajectories.

Reviewer #2 (Remarks to the Author):

This study shows that the endocardial cells in the zebrafish differentiate into hematopoietic cells. Authors used both in vivo high-resolution imaging and photoconversion tracing methods, as well as ex vivo heart culture experiments, to demonstrate the de novo contribution of endocardial cells to hematopoietic cell lineages. Mechanistically, integrin alpha 4 and Vcam1 regulates the attachment of these hematopoietic cells to endocardium. This work is important to the field of hematopoiesis, as endocardial contribution to HSC is a contentious topic with heated debate in the past few years. Elucidation of this issue is critical to our understanding of hematopoiesis in the development and also potential role in cardiac development/valve formation. The study is well-down and major conclusion is supported by the most data. I have a few concerns and questions for authors to clarify.

We thank the reviewer for their highly supportive comments.

This study showed that the hematopoietic-promoting genes are enriched in non-valve endocardial cells. However, Shigeta et al., used genetic lineage tracing to demonstrate that hemogenic endocardium is a de novo source of tissue macrophages in the endocardial cushion, which is the primordial part for cardiac valves. Could this be due to difference of species or technical method used for two studies? Any explanation for this difference in heterogenous endocardial population in generating hematopoietic cells?

The observed discrepancy may stem from inherent evolutionary differences between the mouse and zebrafish. The zebrafish endocardium may exhibit greater plasticity and hematopoietic potential than that of mice. In addition, our study takes advantage of the zebrafish as a model for single-cell imaging, and employed live imaging, FACS analysis, and *in vitro* cell culture to find evidence of endocardial-to-hematopoietic transition. It is possible that genetic lineage tracing in the mouse study may not have detected minor or transient contributions of endocardial cells to hematopoiesis that were observed in the zebrafish through direct visualization methods. However, the absence of Cre-based lineage tracing tools in zebrafish precludes the possibility of a long-term longitudinal study of the persistence, final locations, and functions of endocardial-derived hematopoietic cells. Further comparative studies assessing the hematopoietic potential of endocardial cells across different vertebrate models are necessary to resolve

this apparent contradiction.

It is shown that the majority of HSPC were generated in the outer surface part of the ventricle close to the outflow canal. Any possible explanation for why such regional enrichment of endocardium-derived HSPC?

We have now included new data using RNAscope *in situ* hybridization to detect the expression of *vcam1b*, the critical adhesion molecule known to be enriched in hematopoietic niche endothelial cells and important in homing of HSPCs. *vcam1b* expression appeared enriched in the endocardial cells in the outer curvature of the ventricle, suggesting that these cells provide a more adherent population compared with the rest of the endocardial cells. In contrast, *vcam1b* appeared undetectable in the cardiac valve, where hematopoietic cells were not found. These results are now included in new Figure 5B.

In addition to *vcam1b* expression, cells within the outer curvature of the ventricle or the outflow tract might provide hematopoietic cells with signaling molecules that promote HSPC attachment or proliferation, which we are currently investigating.

Recent study using conventional Cre-loxP genetic lineage tracing showed that endocardial cells minimally contribute to cardiac macrophages and hematopoietic cells in the circulation (Liu et al., JCB 2022). Additionally, intercellular genetic tracing demonstrates no contribution of endocardial cells to hematopoietic cells in the developing heart (Liu et al., Dev Cell 2023). These studies argue against the presence of hematopoietic endothelial cell population in the developing endocardium. Here the evidence of endocardial cells to generate HSPC in zebrafish is contradictory to these recent findings in mouse. Is this due to difference between species or approaches for studying cell lineages among different studies?

The zebrafish studies relied mostly on confocal imaging for direct visualization, photoconversion experiments and genetic mutant/morphant assays, whereas the mouse studies used genetic lineage tracing techniques. It is possible that the endocardial-to-hematopoietic transition observed in zebrafish is a unique feature of teleost or that the mouse studies missed a minor contribution of endocardial cells to hematopoiesis that could be detected in zebrafish. Further research comparing the two models more directly would be needed to resolve this apparent contradiction.

In *Itga4* or *Vcam1b* MO zebrafish, could EHT in the dorsal aorta also be impaired in addition to reduced attachment of hematopoietic cells to endocardium?

The *itga4* and *vcam1b* morphant and mutant phenotypes on EHT were initially reported in Li et al. (2018). In the paper, they found that the numbers of HSPCs, as marked by *runx1* expression and *kdrl*:Dendra tracing, in the DA were comparable with the control, but were significantly lower in the CHT (Figure R4). Thus, they concluded that the EHT in the DA is not affected upon loss of *itga4* or *vcam1b*.

[redacted]

Figure R4. Early EHT in the dorsal aorta is not affected in *itga4*^{cas005} mutants. Adapted from Extended Data Fig. 1b, c, and Figure 1c in Li et al. (2018). Whole-mount *in situ* hybridization shows no change in endothelial (*kdrl*) and HSPC (*runx1*) expression in the dorsal aorta at 36 hpf. Quantification of HSPCs in the dorsal aorta/AGM region of *itga4* mutants shows no significant difference compared with control, whereas HSPCs in the CHT were significantly reduced in *itga4* mutants.

Is there any defect in valve formation in the mutant?

We performed spinning disk imaging of the beating heart in *itga4* and *vcam1b* mutants, and found no valve dysfunction or retrograde blood flow in these animals compared with control (Videos 5-7).

Reviewer #3 (Remarks to the Author):

Comments for Authors

The study performed by Bornhorst et al was aimed, using the zebrafish embryo/larva model, at addressing the potential differentiation of the heart endothelium (the endocardial tissue) into hematopoietic stem and progenitor cells (HSPCs) and of the endocardium to serve as a specific niche to ensure the maintenance of HSPCs. To identify molecular signatures of putative HSPCs lodging in the developing heart, they performed single-cell RNAseq using previously established double transgenic zebrafish lines expressing fluorescent reporters into vascular and hematopoietic cells and identified a population of HSPC candidate cells. This population appears to express markers of well characterized aorta-derived HSPCs generated via endothelial-to-hematopoietic transition (including *gata2b*, *runx1* and *myb*) and to more differentiated progenies (including *mpx* and *mpeg1.1* that are more specific of differentiated myeloid progenies, as well as *gata1* and erythrocyte functional markers). Using photoconversion of the protein Kaede expressed in the entire vascular system, they address the origin of heart HSPCs, after photoconversion of either the entire endocardium, or the dorsal aorta (the site of HSC production). To address the autonomous potency of the heart to produce HSPCs, the authors also cultivated the entire organ in vitro for approximately two days. They then perform functional studies aimed at identifying essential adhesion molecules involved in HSPCs maintenance and interactions with heart niche cells, namely and respectively the integrin *itga4* and the cell-adhesion molecule *vcam1b*. Finally, they provide evidence for the possible compensation by heart HSPCs of the megakaryocyte-erythroid specific lineage (MEPs) of red-blood cell (RBCs) loss after the impairment of RBC production by primitive erythropoiesis, by knocking down the transcription factor *gata1*.

The contribution of the developing heart in the de novo formation of HSPCs remains a debated question in vertebrate models (as mentioned in the authors introduction, although recent evidence in the mouse speaks against immune and blood cells being derived from the heart endothelium, see the matter arising by Liu et al Dev Cell 2023, Ref 34 in the authors ref list).

To tackle this important issue, Bornhorst et al use the power of the zebrafish embryo model to trace, in the entire animal, cell populations after photoconversion (here, for example, from the heart to the first hematopoietic niche (the caudal hematopoietic tissue – CHT) and from

the dorsal aorta (the site of the endothelial-to-hematopoietic transition) to the heart, the latter for questioning the fact that heart HSPCs would only derive from the dorsal aorta). While photoconversion allows, in theory, to precisely address the organ/tissue/single-cell from which a given cell population originates, it requires a very precise methodology for accuracy. In addition, precise delimitation of specific tissue territories to photoconvert needs to be applied to prevent biases in interpretation. These two points are critical and, based on that, I believe that the work performed by Bornhorst et al did not reach the status of providing solid results in favor of de novo formation of HSPCs from the heart endothelium (see the details below). However, the data illustrating the idea of the heart endothelium being a specific niche for HSPC sub-types, rather leading to megakaryocyte-erythroid precursors (MEP), are more convincing (although requiring support with supplementary information on knocking down experiments using morpholinos).

We thank the reviewer for their critical but constructive comments.

Comments on Figures and Text:

1- Figure 1 and S1.

- The genes that characterize cluster 5; more genes should be given (and perhaps an Excel file as Supplemental with top 10-20 top genes at least for all clusters); looking at Suppl. Fig1B, the heat map is not informative, in particular for cluster 5 for which only cytochrome c oxidase (cox) subunits are listed, which are mitochondrial genes (and in general in sc-RNAseq is an indicator of dying cells); can the authors comment on that ?

We have now included an Excel file with the top 50 genes upregulated for all the clusters in Table 1. We have also included an additional subclustering of the endocardial-hematopoietic cluster to discern subpopulations of hematopoietic cells. Regarding cell death, we took multiple steps to ensure that we excluded dead or dying cells in our single-cell sequencing run: during the FACS of endocardial cells, we first used high DAPI signal to label and exclude dead or dying cells from sorting, and during the single-cell RNA sequencing run, we used high mitochondrial content to label and exclude dead or dying cells from further analysis (all cells possessed less than 5% mitochondrial content). Cytochrome c oxidase is not only associated with cell death but also represents the final step of mitochondrial respiration, which leads to oxidative

phosphorylation. These results are consistent with hematopoietic cells undergoing high proliferation rates. This result is also consistent when we analyzed the cell cycle states in which the endocardial-hematopoietic cells are found in, with the majority being in active cell cycle (200 cells in S phase, 83 cells in G2/M phase, 107 quiescent cells in G1 phase). Mutations of *COX* genes also induce cell death in hematopoietic cells¹, affirming their requirements for hematopoietic cell survival. These results are now included in new Figure 1A-B, S1D, S2A-I, and new Table 1.

In addition, the authors need to be careful in the text of the first paragraph of Results and not speak of ‘genes that promote hematopoiesis’ when they refer to enriched genes in cluster 5; *mpx* and *mpeg1.1* are functional markers of neutrophils and macrophages and are expressed in differentiated myeloid lineages. Hence cluster 5 appears to be very heterogeneous, with more or less differentiated cells of myeloid and erythroid lineages (could this cluster be sub-clustered ?). Each cluster as pointed in Suppl. Fig1C should be associated with a given heart-endothelial or hematopoietic cell type population to help the reader for figure analysis. This should be added to Figure 1 and not only to Suppl. Fig1.

Also, Suppl. Fig1D, owing to the importance of *runx1* in hematopoietic populations, the panel should show in which clusters of the UMAP it is enriched, is it in cluster 5 ? and what for *myb* and *drl* ?

We have now renamed the clusters of the single-cell RNA sequencing to the best of our knowledge regarding the enriched genes (Figure 1A). In addition to the endocardial-hematopoietic cluster, we identified the following 6 clusters: valve endocardial cells, valve interstitial cells, quiescent non-valve endocardial cells, dividing non-valve endocardial cells, non-valve endocardial cells in synthesis phase, and endocardial cells exhibiting lymphatic markers. We noted 4 unknown clusters, as we could not discern a clear functional relevance to the enriched genes in these clusters. We have also corrected the wording of “genes that promote hematopoiesis” in describing the endocardial-hematopoietic cluster.

We have now performed the subclustering of the endocardial-hematopoietic cluster to reveal more information about the subtypes of hematopoietic cells isolated from this analysis, which can be further subdivided into 5 subclusters. Out of the 390 cells, we could distinguish two subclusters with enriched platelet (*nfe2*, *mpl*) and HSPC gene expression (165 cells), and a macrophage subcluster (24 cells). A small subset of cells

expresses neutrophil-enriched genes (15 cells). The rest of the cluster expresses high levels of HSPC markers, including *myb*, *runx1*, and *gata2a*. However, we could not clearly distinguish states of HSPC differentiation based on the gene expression of the subclusters. These results are now included in new Figure S2A-I.

- The quantification of co-localization Figure 1H, I (+ Methods); the co-localization of endothelial surface with CD41+ cells (H) and % co-localization (I), this cannot be accurate because endothelial cells are labelled with nls-mCherry that is targeted to the nucleus; hence, contacting surface cannot be determined (because this requires segmentation, hence cell contours). This comment is also valid for Suppl. Figure 4 (B and C).

We have now used *Tg(kdrl:BFP-CAAX)* expression, an endothelial-specific line with membrane-targeted BFP, as the new marker of endothelial cell surfaces. We have now repeated all the imaging and quantification using this line in combination with *Tg(cd41:GFP)* expression. These results have now replaced the old results with *Tg(kdrl:nls-mCherry)* expression, and included in new Figure 1I, J, S4A-C.

We also noted that we made the incorrect conclusion from our previous analyses; the quantification parameter is indicative of the percentage of cell surface areas of HSPCs (*Tg(cd41:GFP)*⁺ cells) that overlapped with the endocardial cells (*Tg(kdrl:BFP-CAAX)* expression). These data would indicate how much contact the HSPCs create with the endocardium, and not the percentage of endocardial surface that is covered with hematopoietic cells. As such, we removed the statement and replaced it with the new conclusion. Our new results show that a greater degree of HSPC surface overlaps with the endocardium from 74 to 98 to 120 hpf, indicating that over time, more contact areas are found between the two cell types. These results suggest that the hematopoietic cells exhibit a greater degree of attachment as development progresses.

To support Figure 1 (in particular panel G but also for a reader less acquainted with heart morphology and 3D shape and connecting veinous/aortic components), a representation of the heart and afferent/efferent vascular systems should be provided (in Supplemental for ex.). This is important in regard to the outflow tract, for reasoning in regard to increase in % positional mapping of CD41+ cells; I should guess it is contacting tissue of aortic nature ? if yes, it is of relevance in regard to photoconversion experiments as well as in regard to heart

dissection for tissue culture (and potential ‘contamination’ with endothelium of aortic nature, see below comments on Figure 2).

We have now included a schematic representation of the heart and vascular systems in our model which is presented in the new Figure 8.

2- Figure 2.

- The photoconversion is not precise. This is exemplified by the panel B that shows photoconversion of the dorsal aorta (DA) at 22hpf; one can clearly see that the bottom parts of inter-segmental vessels have been photoconverted, the aortic wall on the left side of the image only, and a portion of the sub-aortic space. In this situation, the quantitative outcome on HSPCs that will have emerged from the hemogenic endothelium (in the aortic floor) will be minimized in comparison of photoconversion of the whole endocardium (and hence the quantifications in panel F and G make no sense). More precise photoconversion of the hemogenic endothelium (the aortic floor) should be performed. In addition, the reader is missing the information on which region of the DA has been photoconverted (anterior part of the trunk ? entire DA in the trunk ? DA above the CHT region ?). If, as studied in the last part of this paper, part of the heart population of HSPCs that have important physiological functions come from the one or the other of these different regions, more rigorous photoconversion of these different DA regions should be performed.

We have now added extensive experiments to address the reviewer’s comments. We have sub-divided the DA into three regions: the anterior-most region (anterior to the first intersegmental vessel), the middle region above the yolk extension, and the posterior-most region (posterior to the yolk extension until the tail region). We indeed found heterogeneous potential for different DA regions to contribute to the cells in the heart. In particular, the posterior-most and middle regions contribute the most to endocardial-residing cells when imaged 48 hours post conversion (6 and 2.25 cells, respectively, at 74 hpf; 8 and 4.5 cells, respectively, at 98 hpf). The anterior-most DA region contributes almost no cells to the heart (average of 1 cell at 74 hpf and 0.5 cell at 98 hpf). In addition, we also photoconverted the middle and posterior regions of the DA, and found an average of 10 cells residing in the heart, which is a higher number than we previously reported. In our experiments, we narrowed down the regions of

photoconversion to prevent conversion of endothelial cells within the intersegmental vessels. These results are now included in new Figure 4A-M.

Regarding the photoconversion of the endocardium, the reader should see a Z-projection to be able to appreciate precisely the areas that, indeed, have been photoconverted (also, the scales in panels A and B are very different). In addition, the cells that are spotted by the white asterisks are not convincingly positive for photoconverted Kaede (panel C, 74hpf, right). Finally, the CHT is full of pigment cells that can be easily mistaken for fluorescent cells (in the red and green channels); the authors should be cautious (see panel E) and may provide images of non-photoconverted animals. An alternative is to perform spectral deconvolution to eliminate pigment cell background.

We have ensured that pigment cells are not included in our quantitative analysis by matching them with their positions in the CHT using bright-field images (Figure R5). We also always placed the embryos into Ptü water to minimize pigmentation.

Figure R5. Pigmented cells do not hamper Kaede-red/Kaede-green analysis. (A) Representative confocal image of the CHT region labeled with *Tg(fli:Kaede)* and brightfield showing no interference with pigmented cells for analysis. (B) Brightfield image of the CHT region shows no pigment cells covering the HSPC-homing regions. (C) *Tg(fli:Kaede)* image representing Kaede-red/Kaede-green fluorophore localization. Scale bar: 50 µm.

- More precision should be added to the study by performing photoconversion of different areas of the heart endotelium. In link with the positional mapping of the CD41:egfp+ cells Fig 1G and the bias in the region nearby the outflow tract, could that region be more productive for potential de novo HSPC formation ? (if that makes sense for photoconversion at 22 hpf)

We have now added photoconversion experiments that target specific regions of the heart – the atrium, the ventricular outer curvature (VOC), and the ventricular inner curvature (VIC) – and assessed their individual contributions to the other regions of the

heart and CHT. We performed this experiment at 48 hpf, as this timepoint is the period when we can reliably distinguish and photoconvert the different regions of the heart. The looping and ballooning process that take place immediately prior to this timepoint help to clearly distinguish the cardiac regions, particularly the ventricular outer and inner curvatures, using brightfield imaging that marks the entire heart and *Tg(fli1:Kaede)⁺* marker that labels the endocardium. We found that in addition to the VOC, the atrium contributes a significant proportion of cells that extrude and attach to the VOC and the CHT at later timepoints. Thus, we narrowed down the cardiac regions that contribute to circulating hematopoietic cells as the atrium and the VOC. These results have now been included in new Figures 2G-J, S7A-H, S8A-E.

During our photoconversion of the anterior-most portion of DA endothelial cells at 22 hpf, we also photoconverted the presumptive outflow tract region. This portion of the DA contributes to almost no cells in the heart outside of the outflow tract, indicating that these endothelial cells do not contribute to endocardial-residing hematopoietic cells (new Figure 4A-C'; photoconverted outflow tract evident in Figure 4C').

- Heart dissection and in vitro culture is a nice experiment to address the autonomous production of HSPCs by the endocardium. However, addressing this point by using the CD41:eGFP fish line may be problematic because the level of eGFP is very variable in this line (not to speak of mosaicism which is always at risk). Hence, there may be HSPCs that are not detectable at 48hpf and become visible at 74hpf.

In addition, it is difficult to appreciate if part of connecting arterial tissue may have been dissected together with the 48hpf heart; is that possible? also, if the heart is dissected, it may respond to this physiological insult by awaking arterial functions, could this happen?

Probably, a more detailed analysis, using in situ hybridization for example, of the hemogenic status of the heart/heart endothelium should be performed (this addresses endogenous gene expression which prevents from fluorescent Tg line-induced biases. Example of genes that could be tested: *gata2b*, *runx1*).

We have now repeated the *ex vivo* experiments with *Tg(Runx1:mCherry)⁺* hearts, which is less variable and expressed specifically in HSPCs than *Tg(cd41:GFP)* expression. We confirmed that similar to *Tg(cd41:GFP)⁺* cells, *Tg(Runx1:mCherry)⁺* cells were almost absent at 48 hpf when we dissected the hearts, but were prevalent in the hearts that had

been cultured for 24 hours. 15/19 hearts have *Tg(Runx1:mCherry)*⁺ cells, for an average of 2.26 cells per heart. We did not take any major arterial tissues upon cardiac dissection (Figure R6). These results have now been included in new Figures 3A-C.

Although we appreciate that whole mount *in situ* hybridization (WISH) of HSPC genes in extracted hearts would be good evidence, we encountered a technical problem with the extracted hearts that had been cultured for 24 hours being too fragile for WISH.

Figure R6. Representative image of a dissected hearts. (A) Brightfield image showing an extracted heart without any extra non-cardiac tissues, such as arterial tissue. (B) *Tg(kdr1:GFP)* image of a dissected heart fluorescently labeling the endocardium. Scale bar: 40 μ m.

3- Figure 3

The characterization of *itga4* and *vcam1b* morpholinos: the authors should provide data showing the efficiency of the MO at the mRNA or protein levels. Images of embryo/larva should also be provided to show that they can be compared with wild type animals.

The results with the *itga4* mutant line: the effect looks very drastic on the heart morphology, even for heterozygous (panel E, left, eGFP signals for heart endothelial cells). In addition, the *runx1:mCherry* labelling is very odd (above the white asterisks), what are we looking at ?

We have now included the images of the larvae of *itga4* and *vcam1b* morphants and mutants, all of which have comparable body length and morphology (new Figure S9). In addition, we have added endocardial attachment data in *vcam1b* mutants, and included *Runx1:mCherry*⁺ cell quantification that showed consistent results between the mutants and morphants (Figure 5C-F). The *itga4* and *vcam1b* morpholinos have been published and used as loss-of-function tools, and we used the published concentrations in our study². Considering that we also include *itga4* and *vcam1b* mutants, and that the reduction of HSPCs to the endocardium outcomes are consistent between the morphants and the mutants, we conclude that the morpholinos were suitable loss-of-

function tools for these genes. These results have now been included in new Figure 5A-M.

4- The argument on mechanical force and its incidence on endocardial cell formation, as well as attachment (page 8). The text is somehow confusing because jumping from referring to attachment of HSPCs to the endocardium (not impaired by hypo-trabeculation, hence stronger shear stress, nor by biomechanical forces from flow) to endocardial hematopoietic cell formation (that hypothetically also does not require mechanical force). This is a major issue of the paper: the part that the authors attribute to HSPC formation from the endocardium versus attachment (niche aspects).

What would be the vision of the authors on the contribution of *runx1* in this context ? (whose expression is dependent on blood flow, in the context of the endothelial-to-hematopoietic transition in the aortic floor)

We agree with Reviewer 3 that our current data do not conclusively point to a lack of EHT in the endocardium when cardiac contraction is reduced, and as such, rephrased it as attachment of hematopoietic cells to the endocardium being at least in part independent of cardiac contraction and blood flow.

We acknowledge the limitation of this experiment in that we could not completely abolish heartbeat without severely affecting cardiac morphology and inducing potential secondary defects in HSPC attachment to the heart. As such, our conclusions were derived from a reduction (by 30%) and not complete loss of heartbeat as previously reported in other studies in the DA (Figure S11A-C). Nonetheless, we still observed no significant changes in the numbers of HSPCs attached to the endocardium, suggesting a partial independence from contraction/flow-induced biomechanical forces. However, we speculate that cardiac contraction and blood flow, and *runx1* expression, still play a role in the initial EHT of endocardial cells, but not in the later phase of HSPC homing and attachment to the endocardial tissue.

In conclusion, this paper reports more on the attachment/residence of HSPCs and the relationship to the heart cell niche than the formation de novo of HSPCs from the endocardium.

We agree with the reviewer (hence the title of the paper) and have emphasized the role of the heart as a resident tissue instead of a major contributor of hematopoietic cells. We have now added an extra paragraph about the role of endothelial cells in hematopoietic niches in the Introduction and more statements in the Discussion regarding the role of the heart as a resident tissue and potential niche for HSPCs/MEPs.

References

1. Tarasenko TN, Pacheco SE, Koenig MK, et al. Cytochrome c Oxidase Activity Is a Metabolic Checkpoint that Regulates Cell Fate Decisions During T Cell Activation and Differentiation. *Cell Metab.* 2017;25(6):1254-1268.e7. doi:<https://doi.org/10.1016/j.cmet.2017.05.007>
2. Li D, Xue W, Li M, et al. VCAM-1+ macrophages guide the homing of HSPCs to a vascular niche. *Nature.* 2018;564(7734):119-124. doi:10.1038/s41586-018-0709-7

REVIEWERS' COMMENTS

Reviewer #1 (Remarks to the Author):

The authors have convincingly addressed my main concerns. Additionally, they describe the limitations of their approach in the discussion, which is important. I would encourage the authors to include a brief description in the discussion of how HSPCs can be biased towards erythroid or myeloid lineages.

For example, the authors should discuss whether AGM-derived HSPCs reside in the heart after delamination and become erythroid/myeloid biased there. Another scenario is that HSPCs are born erythroid/myeloid biased in the AGM and then migrate to the heart for proliferation. A third possibility is that AGM-derived HSPCs are not lineage biased, but the EdC-derived HSPCs are. Therefore, some HSPCs in the heart may convert to erythroid/myeloid lineages while others may differentiate into all lineages.

Even if these points cannot be proved experimentally with current technology, it would be beneficial to offer a prospective view of the work ahead that needs to be done.

Reviewer #2 (Remarks to the Author):

Authors have addressed my questions with experiments and some explanations.

Reviewer #3 (Remarks to the Author):

Bornhorst et al have performed an extensive revision of their work and answered satisfactorily to most of my comments.

Here are few additional comments to help the authors to finalize their manuscript:

The authors complemented their initial work and identified, after photoconversion, the specific regions of the endocardium that are more prone to provide hematopoietic cells, i.e the atrium and the ventricular outer curvature. As explained by the authors in the point-by-point answers, these experiments were legitimately performed at 48 hpf because it appears to be more appropriate to distinguish unambiguously specific regions of the heart (particularly the ventricular outer and inner curvatures). One should keep in mind however that photoconversion at 48 hpf is always at risk because cells produced by the DA and homing in the heart may become photoconverted. In these conditions, the data shown in the new Figure 3 are more significant for arguing in favor of hematopoietic cells emerging from the endocardium.

For the new Figure 3, the authors have performed tricky experiments, including ex-vivo 24hrs heart cultures and 3D light-sheet microscopy. Heart cultures now show the double labelling of putative heart-derived hematopoietic cells expressing CD41 and Runx1, which is an improvement in comparison to initial data. Also, live imaging with 3D resolution using light-sheet show few double

positive *kdr*/CD41 cells appearing to emerge from the endocardium.

For the new Figure 4 using photoconversion (which is very nice) to address the respective contribution of different aortic segments to produce HSPCs that will colonize the heart, the difference between 'posterior DA' and 'middle and posterior DA' (Fig.4, panels G and J, respectively) is not so clear. Looking at the photoconverted area does not look different between the 2 panels. I advise the authors to check their images (both show the CHT region on the right side (easily recognizable owing to the vein plexus beneath the aorta), and a large part of the aortic segment in the trunk region (the AGM, along the elongated yolk)). In my initial comments, I suggested 'anterior part of the trunk' (which indeed was photoconverted by the authors (including the highest part ahead of the first ISVs), showing that extremely little to virtually no cells from that region colonize the heart), 'the entire DA in the trunk' (meaning the DA along the elongated yolk, i.e in the AGM region), 'the aorta above the CHT region'. I believe being very precise here is a relatively important point because the aorta in the trunk region provides HSPCs with longer regenerative potential than for the aorta in the tail (above the CHT), as was shown in Z. Wen's lab (see Tian et al., 2017 PMID: 28931624); this may be an issue for producing cells with more or less long term regenerative potential and becoming - or not -, resident in the heart.

For the scRNAseq data of cluster 5, notice is taken that the hematopoietic genes (*myb*, *drl*, *drll*, *hdr*, *ikzf*, *gata2b*) do not appear in the top 30, if I am correct looking at Table 1 (with only *hdr* in the top 50; however, as show Fig.1C and to some extent Fig.1S, they appear to be enriched in cluster 5 in comparison to other clusters).

Finally, the authors have taken care to mention (in Introduction and Result sections) the recent article by the Sumanas lab (Gurung et al 2024, Ref.49). In this work, also describing the emergence of hematopoietic precursor cells from the heart endothelium, scRNAseq experiments show that these cells are more specifically leading to neutrophils. Perhaps the authors should emphasize that these analyses were performed at 24 hpf (in the Results section), which is earlier than in their conditions (48 and 72 hpf), and mention that this may explain the difference.

REVIEWERS' COMMENTS

We thank all reviewers for their highly constructive feedback and final positive comments on our manuscript.

Reviewer #1 (Remarks to the Author):

The authors have convincingly addressed my main concerns. Additionally, they describe the limitations of their approach in the discussion, which is important. I would encourage the authors to include a brief description in the discussion of how HSPCs can be biased towards erythroid or myeloid lineages.

For example, the authors should discuss whether AGM-derived HSPCs reside in the heart after delamination and become erythroid/myeloid biased there. Another scenario is that HSPCs are born erythroid/myeloid biased in the AGM and then migrate to the heart for proliferation. A third possibility is that AGM-derived HSPCs are not lineage biased, but the EdC-derived HSPCs are. Therefore, some HSPCs in the heart may convert to erythroid/myeloid lineages while others may differentiate into all lineages.

Even if these points cannot be proved experimentally with current technology, it would be beneficial to offer a prospective view of the work ahead that needs to be done.

We thank the reviewer for their positive comments and have expanded on the discussion segment in the second paragraph in the Discussion on how cardiac-residing HSPCs might be biased towards a specific differentiated lineage (erythrocyte/platelet lineage).

Reviewer #2 (Remarks to the Author):

Authors have addressed my questions with experiments and some explanations.

We thank the reviewer for their positive feedback.

Reviewer #3 (Remarks to the Author):

Bornhorst et al have performed an extensive revision of their work and answered satisfactorily to most of my comments.

We thank the reviewer for their positive feedback.

Here are few additional comments to help the authors to finalize their manuscript:

The authors complemented their initial work and identified, after photoconversion, the specific regions of the endocardium that are more prone to provide hematopoietic cells, i.e the atrium and the ventricular outer curvature. As explained by the authors in the point-by-point answers, these experiments were legitimately performed at 48 hpf because it appears to be more appropriate to distinguish unambiguously specific regions of the heart (particularly the ventricular outer and inner curvatures). One should keep in mind however that photoconversion at 48 hpf is always at risk because cells produced by the DA and homing in the heart may become photoconverted. In these conditions, the data shown in the new Figure 3 are more significant for arguing in favor of hematopoietic cells emerging from the endocardium.

For the new Figure 3, the authors have performed tricky experiments, including ex-vivo 24hrs heart cultures and 3D light-sheet microscopy. Heart cultures now show the double labelling of putative heart-derived hematopoietic cells expressing CD41 and Runx1, which is an improvement in comparison to initial data. Also, live imaging with 3D resolution using light-sheet show few double positive *kdr1*/CD41 cells appearing to emerge from the endocardium.

We thank the reviewer for their positive comment.

For the new Figure 4 using photoconversion (which is very nice) to address the respective contribution of different aortic segments to produce HSPCs that will colonize the heart, the difference between ‘posterior DA’ and ‘middle and posterior DA’ (Fig.4, panels G and J, respectively) is not so clear. Looking at the photoconverted area does not look different between the 2 panels. I advise the authors to check their images (both show the CHT region on the right side (easily recognizable owing to the vein plexus beneath the aorta), and a large part of the aortic segment in the trunk region (the AGM, along the elongated yolk)). In my initial comments, I suggested ‘anterior part of the trunk’ (which indeed was photoconverted by the authors (including the highest part ahead of the first ISVs), showing that extremely little to virtually no cells from that region colonize the heart), ‘the entire DA in the trunk’ (meaning the DA along the elongated yolk, i.e in the AGM region), ‘the aorta above the CHT region’. I believe being very precise here is a relatively important point because the aorta in the trunk region provides HSPCs with longer regenerative potential than for the aorta in the tail (above the CHT), as was shown in Z. Wen’s lab (see Tian et al., 2017 PMID: 28931624); this may be an issue for producing cells with more or less long term regenerative potential and becoming - or not -, resident in the heart.

We thank the reviewer for their useful comment concerning the segment differences between ‘posterior DA’ and ‘middle and posterior DA’. Within the DA, we defined the anterior portion as the anterior-most part of the DA until just before the first intersegmental vessel (ISV) sprout (Figure 4A), the “middle region” as the DA region adjacent to the first 12 ISV sprouts (Figure 4D), and the “posterior region” as the

posterior-most DA region from the 13th ISV sprout until the tail (Figure 4G). We have included this classification in the Methods section.

For the scRNAseq data of cluster 5, notice is taken that the hematopoietic genes (myb, drl, drll, hdr, ikzf, gata2b) do not appear in the top 30, if I am correct looking at Table 1 (with only hdr in the top 50; however, as show Fig.1C and to some extent Fig.1S, they appear to be enriched in cluster 5 in comparison to other clusters).

Indeed, these genes are enriched in Cluster 5 as seen in Fig. 1C and Supplemental Figure 1.

Finally, the authors have taken care to mention (in Introduction and Result sections) the recent article by the Sumanas lab (Gurung et al 2024, Ref.49). In this work, also describing the emergence of hematopoietic precursor cells from the heart endothelium, scRNAseq experiments show that these cells are more specifically leading to neutrophils. Perhaps the authors should emphasize that these analyses were performed at 24 hpf (in the Results section), which is earlier than in their conditions (48 and 72 hpf), and mention that this may explain the difference.

We have included this statement in the Results section of the manuscript.